# Epidermal chloroplasts are defense-related motile organelles equipped with plant immune components

Hiroki Irieda [1✉] & Yoshitaka Takano[2]

In addition to conspicuous large mesophyll chloroplasts, where most photosynthesis occurs, small epidermal chloroplasts have also been observed in plant leaves. However, the functional significance of this small organelle remains unclear. Here, we present evidence that *Arabidopsis* epidermal chloroplasts control the entry of fungal pathogens. In entry trials, specialized fungal cells called appressoria triggered dynamic movement of epidermal chloroplasts. This movement is controlled by common regulators of mesophyll chloroplast photorelocation movement, designated as the epidermal chloroplast response (ECR). The ECR occurs when the PEN2 myrosinase-related higher-layer antifungal system becomes ineffective, and blockage of the distinct steps of the ECR commonly decreases preinvasive nonhost resistance against fungi. Furthermore, immune components were preferentially localized to epidermal chloroplasts, contributing to antifungal nonhost resistance in the *pen2* background. Our findings reveal that atypical small chloroplasts act as defense-related motile organelles by specifically positioning immune components in the plant epidermis, which is the first site of contact between the plant and pathogens. Thus, this work deepens our understanding of the functions of epidermal chloroplasts.

[1] Academic Assembly, Institute of Agriculture, Shinshu University, Nagano, Japan. [2] Graduate School of Agriculture, Kyoto University, Kyoto, Japan.
✉email: irieda@shinshu-u.ac.jp

The epidermis of multicellular organisms acts as a boundary with the environment to protect against a wide variety of biotic and abiotic stresses. Although the plant epidermis is the site of primary pathogen recognition, it can also be attacked by pathogens, resulting in infection. For instance, the anthracnose fungi *Colletotrichum* species and the rice blast fungus *Magnaporthe oryzae* develop a specialized dome-shaped fungal cell called an appressorium, which is highly pigmented with melanin and is used in direct fungal entry into plant epidermis[1]. The melanized appressorium produces a penetration peg, followed by the extension of the invasive hyphae. However, if the plant is a nonhost, a robust and broad-spectrum defense response termed nonhost resistance (NHR) is developed to effectively block the infection of a vast number of nonadapted pathogens[2]. More specifically, epidermal immune responses are of great importance in terminating the entry of nonadapted fungal pathogens.

PEN2 myrosinase is known as a positive regulator of epidermal NHR in *Arabidopsis thaliana* against multiple fungal pathogens, including *Blumeria graminis* f. sp. *hordei*, *Colletotrichum tropicale* (*Ctro*), and *Magnaporthe oryzae*[3–7]. Together with the ATP binding cassette transporter PEN3, PEN2 regulates the tryptophan-derived secondary metabolite-mediated defense pathway[3–5,8,9]. In addition, EDR1 protein kinase has a positive effect on the preinvasive NHR to nonadapted *Ctro*, whereas it has negative effects on immunity to the adapted fungus *Golovinomyces cichoracearum*[10,11]. However, *pen2* and *edr1* mutants retain normal resistance against the nonadapted fungus *Colletotrichum orbiculare* 104-T (*Corb*)[7,11]. Therefore, multiple immune components, including as-yet unknown factors, support the deployment of epidermal NHR against the entry of nonadapted fungal pathogens in *Arabidopsis*.

In plant immunity, chloroplasts play an important role in the biosynthesis of defense-related molecules[12]. Secondary messengers such as reactive oxygen species (ROS) and calcium ($Ca^{2+}$) are produced and released by chloroplasts, respectively; precursors of phytohormones such as salicylic acid (SA), jasmonic acid, and abscisic acid are also synthesized in chloroplasts[13–16]. It has been proposed that immune signals triggered by pathogen-associated molecular patterns (PAMPs) are transmitted from the cell surface to chloroplasts via intracellular $Ca^{2+}$ relay, followed by chloroplast-to-nucleus retrograde ROS signaling in *A. thaliana*[17]. In chloroplast-to-nucleus ROS signaling in *Nicotiana benthamiana* in response to viral and bacterial pathogens, the involvement of tubular extensions from the chloroplast (called stromules) has been proposed[18].

In higher plants, it has been widely recognized that no chloroplasts exist in epidermal cells other than guard cells, with the exception of some plants (e.g., tobacco)[19–22]. A large population of chloroplasts is highly differentiated in leaf mesophyll cells for efficient photosynthesis, while guard cell chloroplasts regulate stomatal movements in response to $CO_2$ and light[23]. However, chlorophyll-containing chloroplasts in the epidermal pavement cells of *A. thaliana* have been recognized, although their size and number are smaller than those of mesophyll cells[24,25]. The number and thickness of grana and the chlorophyll content of epidermal chloroplasts are also low, leading to the plausible assumption that the photosynthetic contribution of these small organelles is lower than that of mesophyll cells[24]. Hence, the functional significance of small epidermal chloroplasts remains unclear.

Here, we report that epidermal chloroplasts dynamically respond to *Colletotrichum* and *Magnaporthe* fungi that are not adapted to *Arabidopsis*. We showed that epidermal chloroplasts emerge at the upper periclinal wall in response to the entry trial of these nonadapted fungi (hereafter referred to as the epidermal chloroplast response, ECR). The frequency of the ECR increased

in the absence of the PEN2-based antifungal defense system, suggesting that the ECR is newly activated when PEN2-related defense becomes ineffective. Our data revealed that CHUP1 and JAC1, regulators of chloroplast photorelocation movement in mesophyll cells, also control the ECR. Importantly, blocking the distinct steps of the ECR via mutation or overexpression of the *CHUP1* gene significantly reduced preinvasive NHR against nonadapted fungi in epidermal cells. Furthermore, we found that multiple immune components that are preferentially localized to epidermal chloroplasts contribute to preinvasive NHR in the lower layer of PEN2-related immunity. We propose that atypical small chloroplasts in the epidermis function as defense-related motile organelles involved in *Arabidopsis* NHR.

## Results

**Epidermal chloroplasts emerged on the upper periclinal surface in response to the entry trial by nonadapted *Colletotrichum* fungi.** To investigate the relationship between epidermal chloroplasts and preinvasive NHR against the appressorium-mediated aggression of fungal pathogens in *A. thaliana*, we used three nonadapted *Colletotrichum* strains: *C. fioriniae* CC1 (*Cfio*), *C. siamense* MAF1 (*Csia*), and *Corb*, which are cosmos, apple, and cucurbit anthracnose fungi, respectively. We also used the adapted strain *C. higginsianum* Abr1-5 (*Chig*), a brassica anthracnose fungus. *Cfio*, *Csia*, or *Corb* did not form any lesions in wild-type *Arabidopsis* (Col-0), whereas *Chig* developed severe lesions (Fig. 1a). Interestingly, microscopic observation revealed that chlorophyll-containing epidermal chloroplasts emerged on the upper periclinal wall (surface) of pavement cells after inoculation with nonadapted *Colletotrichum* fungi, especially *Cfio* and *Csia* (Fig. 1b, c). The degree of this response varied according to the fungal strain, but the response was detected at 1 day post-inoculation (dpi) and tended to increase until 3 dpi. This phenomenon was also observed in a transgenic plant line expressing the plastid-cyan fluorescent protein (CFP) marker (Supplementary Fig. 1). We named this phenomenon the ECR, because the epidermal chloroplasts are usually positioned at the lower periclinal (bottom) and anticlinal walls in a steady state. The ECR against *Chig* was not clearly detectable, suggesting that the ECR is a specific response to nonadapted fungi (Fig. 1c).

Since *Corb* activated the ECR on Col-0 at a lower level compared with *Cfio* and *Csia*, we hypothesized that some preinvasive defenses in NHR may take priority over the ECR. Consistent with this idea, we found that *pen2* plants showed increased ECR against *Cfio*, *Csia*, and *Corb* compared to Col-0 plants (Fig. 1c). This suggests that the ECR occurs preferentially in the absence of PEN2-related immunity. Indeed, *pen2* increased epidermal invasion (at 4 dpi) and leaf lesion formation (at 7 dpi) by *Cfio* and *Csia*, indicating compromised NHR in *pen2* plants (Fig. 1a, d, e). Quantitative real-time PCR (RT-qPCR) analysis also showed that many defense-related genes were induced only in *pen2* at 1 dpi (Supplementary Fig. 2). This suggests the presence of a preinvasive defense system that is activated in the absence of PEN2, further supporting that the PEN2-related pathway is a higher-layer preinvasive defense system against *Colletotrichum* fungi. Remarkably, the impact of *pen2* mutation on *Corb*-induced ECR was drastic, while *Corb* was not able to invade *pen2* (Fig. 1d, e). Since the ECR is not accompanied by fungal invasion, this may imply that the ECR contributes to preinvasive NHR.

Interestingly, we detected only a slight increase in the frequency of the ECR in *edr1* mutants (Fig. 1c). Indeed, unlike the *pen2* mutant, the preinvasive NHR of the *edr1* mutant against appressorium-mediated *Cfio* and *Csia* entry did not decrease, with no induction of defense-related genes (Fig. 1e, Supplementary Fig. 2); this result strengthens the link between the

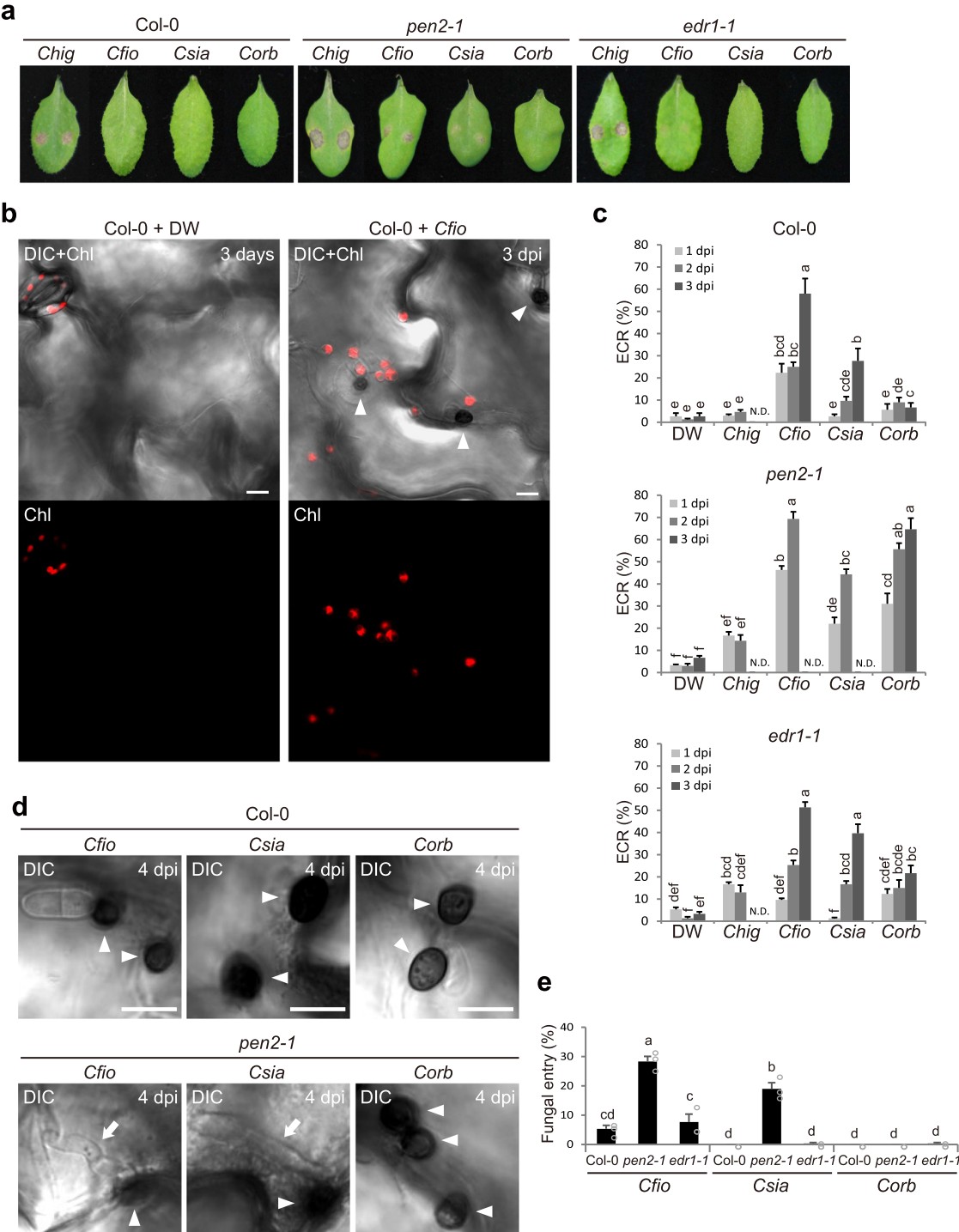

**Fig. 1 The ECR occurs at the lower layer of PEN2-related immunity and is critical for NHR against *Colletotrichum* fungi. a** Pathogenicity of *Colletotrichum* fungi on *Arabidopsis* mutants. A conidial suspension of adapted *Chig* and nonadapted *Cfio*, *Csia*, and *Corb* was inoculated onto leaves of wild-type *Arabidopsis* (Col-0), *pen2-1*, and *edr1-1* and incubated for 7 days. **b** ECR after *Cfio* inoculation. The epidermal surface of the *Cfio*-inoculated cotyledon of Col-0 was investigated at 3 days post-inoculation (dpi). The chloroplasts were visualized based on chlorophyll autofluorescence. The DIC images were captured by confocal microscopy. DW was used as a control. **c** The ratio of epidermal cells with surface chloroplasts was investigated at 1, 2, and 3 dpi. A total of 100 cells in contact with the melanized appressorium were observed. N.D.: not determined due to damages of epidermal cell by fungal invasion. **d**, **e** Fungal invasion of *Arabidopsis* epidermis. *Cfio*, *Csia*, and *Corb* were inoculated onto cotyledons of the indicated *Arabidopsis*. The entry ratio was quantified at 4 dpi. A total of 100 melanized appressoria were investigated. The arrowheads and arrows indicate melanized appressoria and invasive hyphae, respectively. Scale bar, 10 μm. The means and SE were calculated from three independent plants. Means not sharing the same letter are significantly different (*P* < 0.05, two-way ANOVA with Tukey's HSD test).

breakdown of higher-layer preinvasive defenses such as the PEN2-related pathway and ECR activation.

**The ECR was correlated with fungal secretion activity during the entry trial**. We focused on the triggers of the ECR. One candidate trigger is pathogen-derived PAMPs. The immune kinases BAK1, BIK1, and PBL1 function in PAMP-triggered immunity[26–29]. However, *bak1-5* and *bik1 pbl1* mutants showed normal ECR (Supplementary Fig. 3). Thus, we analyzed fungal factors other than PAMPs that may induce the ECR. In *Colletotrichum* fungi, the yeast *STE12* gene homolog *CST1* and tetraspanin gene *PLS1* are involved in appressorium-mediated penetration[30,31]. The *cst1*[30] and *pls1*[31] mutants could form melanized appressoria, but were defective in penetration peg formation. In contrast, the sterol glucosyltransferase gene *ATG26* and isocitrate lyase gene *ICL1* were not involved in penetration peg formation, but were involved in subsequent invasive hyphal development in planta[32,33]; therefore, *atg26*[32] and *icl1*[33] mutants were defective in invasive hyphal development. We first inoculated *pen2* plants with these *Corb* mutants (*pls1* was generated in this study, Supplementary Fig. 4) and found that *atg26* and *icl1* could induce the ECR, while *cst1* and *pls1* could not (Fig. 2a). This suggests that fungal penetration peg formation is essential for triggering the ECR (Fig. 2a). We examined the possibility that plant damage caused by penetration peg formation activated the ECR. The wound-induced endogenous peptide Pep1 and its paralogs are recognized by the damage-associated molecular pattern (DAMP) receptors PEPR1 and PEPR2 in *Arabidopsis*[34,35]. However, we detected normal ECR in *pepr1 pepr2* mutants (Supplementary Fig. 3). Combined with the fact that BAK1 and BIK1 also function in Pep1 DAMP signaling[36,37], Pep1 and its paralogs might be not a cue of ECR.

Next, we examined whether fungal secretions from peg triggered the ECR. We inoculated *pen2* plants with the *Corb* expressing mCherry-labeled effector CoDN3[38] and found that the effector signal at the penetration pore was stronger than that on Col-0 (Fig. 2b). Similar results were observed on *pen3*, implying the possible involvement of tryptophan-derived secondary metabolites in suppressing fungal effector delivery. A gene disruption mutant of the membrane traffic component *SEC22* in *Corb*, Δ*sec22*, showed decreased effector secretion, and the effector signals were abnormally retained and dispersed inside fungal cells when Δ*sec22* was inoculated on host cucumber[38]. As expected, the focal accumulation of the effector at the penetration pore, located in the center at the bottom of the appressorium, was attenuated in Δ*sec22* when Δ*sec22* was inoculated on nonhost *Arabidopsis* (Fig. 2b). Additionally, the effector exhibited abnormal retention inside the appressorium (Fig. 2b). Intriguingly, the ECR significantly correlated with fungal secretion activity at the penetration pore (Fig. 2b, c), whereas the *Corb* Δ*sec22* mutant induced sufficient papillary callose deposition at the entry trial sites of the fungus; papillary callose deposition is one of the defense responses of the plant against fungal entry trials (Fig. 2d). Since penetration peg-defective *cst1* and *pls1* mutants hardly induced papilla (Fig. 2e), these results imply that *A. thaliana* recognizes pathogen-derived secretions from the peg and activates the ECR. However, there is still a possibility that the slight differences in the degree of fungal progression in the penetration attempt, which is not reflected by papilla formation, affects ECR activation.

**The ECR shares common regulators with mesophyll chloroplast photorelocation movement**. In mesophyll cells, typical large chloroplasts exhibit photorelocation movement[39]. Under weak light, they move toward the periclinal walls to promote photosynthesis (accumulation response). Under strong light, they migrate to the anticlinal wall to reduce photodamage (avoidance response). CHUP1 generates a chloroplast-actin-based motive force through the polymerization of chloroplast-actin; therefore, CHUP1 is essential for both accumulation and avoidance responses[40–42]. On the other hand, JAC1 plays a role in the appearance and disappearance of chloroplast-actin filaments and is required for the accumulation response[41].

To gain new insight into the relationship between these components and the ECR, we examined how the levels of CHUP1 and JAC1 proteins affected the ECR (Fig. 3). We generated transgenic plants overexpressing *CHUP1* (*CHUP1*ox) or *JAC1* (*HA-JAC1*ox) and analyzed them with *chup1* and *jac1* mutants. The growth of these plants was normal, except that *HA-JAC1*ox exhibited a semi-curly leaf phenotype (Fig. 3a, Supplementary Fig. 5). Intriguingly, CHUP1 overexpression completely suppressed the ECR (Fig. 3b, c). In contrast, the *chup1* mutant constitutively displayed epidermal chloroplasts at the surface with or without fungal inoculation (Fig. 3b, c). An inoculum with high concentration of fungal conidia on the *chup1* mutant did not significantly increase the population of ECR-activated cells, but the number of surface chloroplasts was slightly increased (Fig. 3d). Therefore, the *chup1* mutant showed constitutive positioning of the epidermal chloroplasts at the surface, but the mutant is deficient in ECR that is newly triggered by the inoculation of nonadapted *Colletotrichum* fungi (Fig. 3b–d). Thus, both overexpression and mutation of CHUP1 commonly cause impairments in the intracellular movement of epidermal chloroplasts in the ECR. On the contrary, JAC1 overexpression resulted in a *chup1*-like phenotype, although the *jac1* mutant showed near-normal ECR (Fig. 3b, c).

Mutations in hydrophilic amino acid residues in CHUP1 (R4A&S12A&R20A) cause them to change location from the perichloroplasts to the cytosol[42]. We generated *CHUP1-R4A&S12A&R20A*ox lines and found that this point-mutated CHUP1 exhibited decreased endogenous CHUP1 proteins, thereby resulting in a *chup1*-like dominant negative phenotype; epidermal chloroplasts were constitutively detected at the surface without fungal inoculation, and this phenotype was correlated with the levels of endogenous CHUP1 proteins (Fig. 3a, e, Supplementary Fig. 5). In brief, the dominant negative effects of these mutant lines were partial; therefore, the ECR still occurred in response to Cfio (Fig. 3e). These results further suggest that CHUP1 negatively regulates the ECR.

Chloroplast photorelocation is mediated by the blue light receptors phototropin 1 (phot1) and 2 (phot2)[43–45]. Since phot2 is involved in both accumulation and avoidance responses, we evaluated the ECR of the *phot2* and *phot1 phot2* mutants. We found that the ECR occurred normally, except for a slight increase in surface chloroplasts in *phot1 phot2* without fungal inoculation (Fig. 3f). These findings suggest that chloroplast photorelocation and the ECR have different stimulus recognition systems, although they might share downstream components.

**The ECR contributes to NHR against *Colletotrichum* fungi**. To clarify whether the ECR is involved in plant immunity, we generated *CHUP1*ox and *chup1* plants in the *pen2* mutant background (Fig. 4). The fungal entry rates on these generated plants were higher than those on *pen2*, while the *CHUP1*ox and *chup1* single mutants retained normal resistance (Fig. 4a). Transgenic *pen2 chup1* lines expressing *CHUP1* under its own promoter showed complementation of the ECR and antifungal immunity (Fig. 4b–d). Intriguingly, we found that the *edr1 pen2* double mutant exhibited an increased entry rate compared with the *pen2* single mutant, suggesting that the effect of *edr1* mutation on

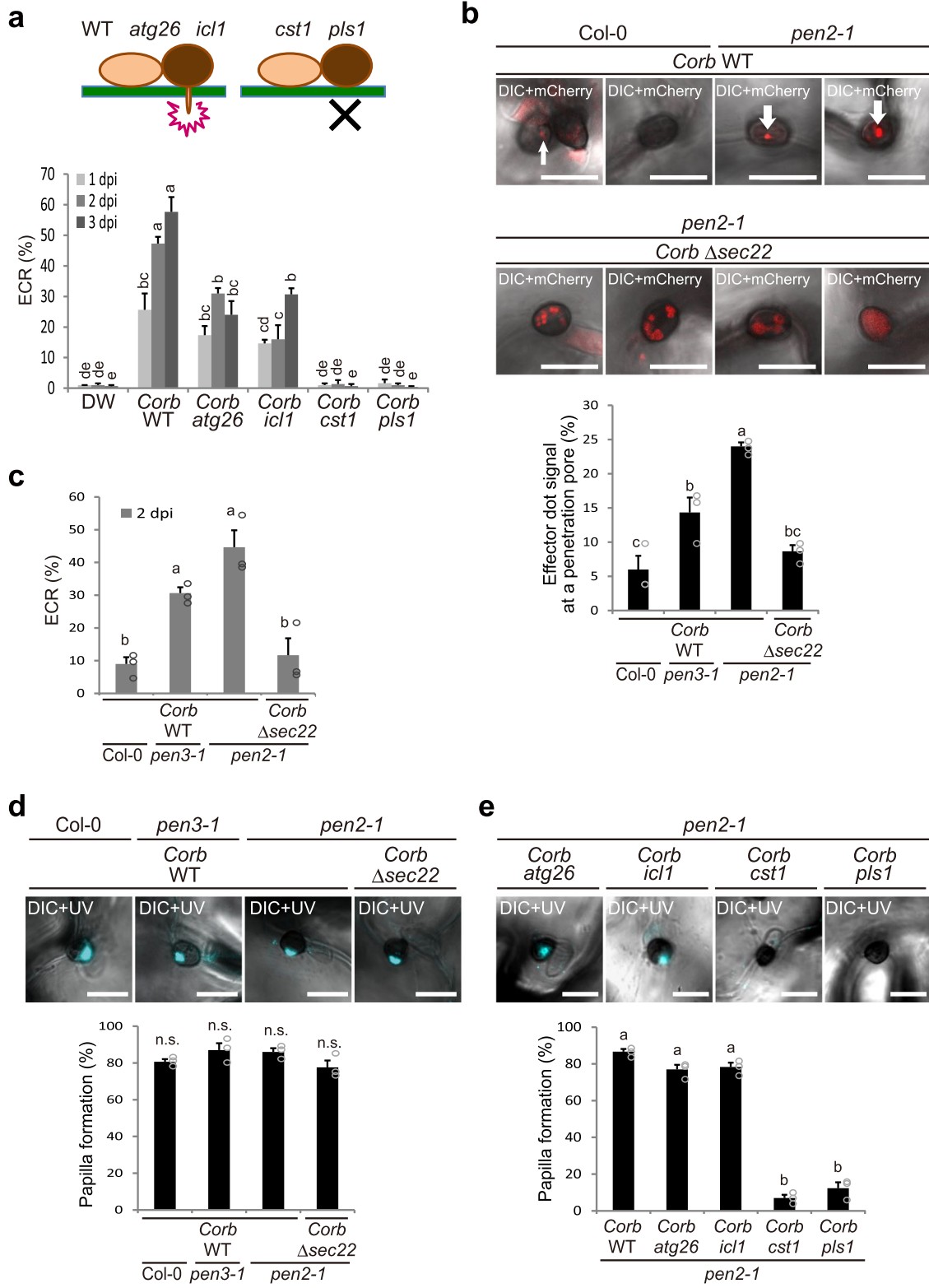

preinvasive NHR became apparent in the *pen2* background (Fig. 4a). Moreover, the effects of ECR impairment by *CHUP1* overexpression and mutation on preinvasive NHR were greater in *edr1 pen2* double mutants than in *pen2* single mutants. These results demonstrate that the ECR contributes to preinvasive NHR against *Colletotrichum* fungi at the lower layer of PEN2-related immunity, while the breakdown of additional EDR1-related preinvasive defense has a greater impact on ECR-mediated

immunity. RT-qPCR analysis showed a significant decrease or decreasing trend in the expression of some *Cfio*-induced genes in *pen2 CHUP1*ox and *pen2 chup1* compared to that in *pen2*, which suggests the possible involvement of the ECR in regulating these defense-related genes (Fig. 4e).

When nonadapted *Colletotrichum* fungi invade *Arabidopsis*, postinvasive NHR responses, including plant cell death, are activated[46]. Therefore, we examined the possible involvement of

**Fig. 2 Fungal peg formation and penetration pore-mediated secretion correlate with plant ECR during entry trial. a** ECR after inoculation of *Corb* mutants. Wild-type (WT) *Corb*, *atg26*, *icl1*, *cst1*, and *pls1* were inoculated onto cotyledons of *pen2-1* plants. The ECR was quantified at 1, 2, and 3 dpi. DW was used as a control. **b** Focal accumulation of *Corb* effector at the penetration pore was abolished by disruption of the *SEC22* gene in *Corb*. *Corb* expressing CoDN3-mCherry was inoculated onto the indicated *Arabidopsis*. At 2 dpi, the images were captured by confocal microscopy, and the ratio of the effector signals at the penetration pore was quantified. The arrows indicate effector accumulation at the penetration pore. **c** ECR after inoculation of *Corb* WT and Δ*sec22* at 2 dpi. **d**, **e** Papilla formation at the entry trial site of *Corb* WT, Δ*sec22*, *atg26*, *icl1*, *cst1*, and *pls1*. *Corb*-inoculated plants were subjected to staining with aniline blue fluorochrome for visualization of callose deposition. At 1 dpi, the images were captured, and the ratio of papillary callose deposition was investigated. For all experiments, 100 melanized appressoria were observed, and the means and SE were calculated from three independent plants. Scale bar, 10 μm. Means not sharing the same letter are significantly different ($P < 0.05$, one-way (**b–e**) or two-way (**a**) ANOVA with Tukey's HSD test). n.s. not significant.

the ECR in postinvasive NHR. Trypan blue (TB) staining indicated the induction of cell death in the epidermal pavement and palisade mesophyll (not invaded) cells inoculated with *Cfio* (Fig. 4f), which is consistent with a previous report[46]. The invasive hyphae were confined to the dead epidermal cell, suggesting that postinvasive defense in ECR-defective plants functioned normally (Fig. 4f).

Several preinvasive NHR components, such as PEN1, PEN2, PEN3, and PEN4/PCS1, are known to localize at pathogen entry trial sites[3,7,8,47,48]. We also detected PEN2-GFP at the entry trial sites of *Cfio*, *Csia*, and *Corb* (Supplementary Fig. 6). However, epidermal chloroplasts did not show specific accumulation at the entry trial sites (Figs. 1b, 3b, Supplementary Fig. 6); instead, we frequently observed rapidly moving chloroplasts on the upper surface (Supplementary Fig. 7, Supplementary Movie 1). Although epidermal chloroplasts were occasionally observed near the appressoria, the majority of the population in the surface chloroplasts was scattered.

**Immune components GSH1, EDS5, and CAS preferentially localize to epidermal chloroplasts and stromules in preinvasive NHR.** The immune components GSH1, EDS5, and CAS have been reported to localize to small epidermal chloroplasts[46,49,50]. In previous reports, EDS5-GFP preferentially localized around epidermal chloroplasts compared to mesophyll chloroplasts when expressed under both its own and cauliflower mosaic virus (CaMV) *35S* promoters[49]. When only expressed under the CaMV *35S* promoter, mesophyll chloroplasts had weak EDS5-GFP signals, but the chloroplast periphery-specific pattern was maintained in two types of chloroplasts[49]. CAS-GFP driven by the CaMV *35S* promoter was detected in both epidermal and mesophyll chloroplasts, with no difference in their localization patterns[50]. The localization of GSH1-GFP driven by its own promoter in the mesophyll has yet to be confirmed[46]. GSH1 γ-glutamylcysteine synthetase contributes to NHR against *Ctro* via glutathione biosynthesis[46,51]. EDS5 is a MATE family transporter that is required for the transport of the SA precursor outside the chloroplasts[52,53]. CAS is a $Ca^{2+}$-sensing receptor involved in transient $Ca^{2+}$ signaling in chloroplasts during plant immunity[17]. To analyze the subcellular localization of these components in ECR-activated cells, we used *gsh1* plants expressing GSH1-GFP[46] and generated transgenic Col-0 plants expressing EDS5-super folder GFP (sfGFP) or CAS-sfGFP under the control of the CaMV *35S* promoter. In a steady state, these three components localized to the small epidermal chloroplasts positioned at the bottom of the epidermal cells (Fig. 5 upper), which is consistent with previous reports[49,50]. At the surface area, we could detect fluorescent signals of these components only on the guard cell chloroplasts. Importantly, after *Cfio* inoculation, GSH1, EDS5, and CAS localized to ECR-activated surface chloroplasts (Fig. 5 lower). With or without *Cfio* inoculation, all the fluorescent signals of these components in small epidermal chloroplasts were much stronger than those in the large mesophyll chloroplasts,

while CAS-sfGFP was also detected in the epidermal plastid with no chlorophyll (Fig. 5). These localization patterns are different from those of CHUP1, which localizes to both epidermal and mesophyll chloroplasts[54]. Z-stack and sectional images clearly showed preferential localization of GSH1, EDS5, and CAS to the small epidermal chloroplasts (Supplementary Fig. 8, Supplementary Movies 2–7). Altogether, we concluded that the immune components GSH1, EDS5, and CAS preferentially localized to small epidermal chloroplasts in preinvasive NHR.

To further strengthen the link between the ECR and epidermal chloroplast-localized GSH1, EDS5, and CAS, we generated fluorescent marker plant lines of these immune components in the *CHUP1*ox and *chup1* backgrounds (Fig. 6). We found that their localization patterns were strongly correlated with epidermal chloroplasts and the ECR (Figs. 5 and 6, Supplementary Movie 8). These observations suggested that the ECR changes the intracellular locations of the immune components GSH1, EDS5, and CAS.

We also detected fluorescent signals of GSH1-GFP, EDS5-sfGFP, and CAS-sfGFP in stromules (Figs. 5 and 6a), where the plastid-CFP marker was observed, but chlorophyll autofluorescence was not (Supplementary Fig. 1). In the immune response of *N. benthamina*, induced stromules are proposed to be a chloroplast-to-nucleus route for ROS signaling and the transport of NRIP1, which is a protein required for the recognition of the tobacco mosaic virus effector p50[18]. *N. benthamiana* NRIP1, which is exogenously expressed in *A. thaliana*, also localizes to stromules[18]. Therefore, the stromules possibly link to the regulation of the localization of *Arabidopsis* immune-related proteins via ECR.

**GSH1, EDS5, and CAS contribute to preinvasive NHR against *Colletotrichum* fungi.** To evaluate the contribution of GSH1, EDS5, and CAS to preinvasive NHR against *Colletotrichum* fungi in epidermal cells, we quantified the entry rates of *Cfio*, *Csia*, and *Corb*. Hiruma et al. reported that *gsh1* mutants showed a substantial decrease in preinvasive defense against appressorium-independent atypical entry of *Ctro*[46]. On the other hand, the *gsh1* mutant showed a slight reduction in epidermal NHR against appressorium-mediated *Cfio* entry, and the effect was weaker than that in *pen2*. Meanwhile, *Csia* and *Corb* barely invaded the *gsh1* mutant (Fig. 7a). The *eds5* and *cas* mutants retained normal resistance against these *Colletotrichum* fungi (Fig. 7a). Furthermore, the *gsh1*, *eds5*, and *cas* mutants exhibited normal ECR (Supplementary Fig. 9). Remarkably, these three mutations significantly reduced the preinvasive NHR against *Cfio* and *Csia* in the *pen2* background (Fig. 7a). In particular, GSH1 was of paramount importance for preventing fungal entry in the *pen2* background. These results demonstrated the contribution of these components to preinvasive NHR and the priority of PEN2-related defense against *Colletotrichum* fungi.

The *Cfio*-induced expression of the defense-related genes *PR1* and *FRK1* in *pen2 gsh1*, *pen2 eds5*, and *pen2 cas* double mutants

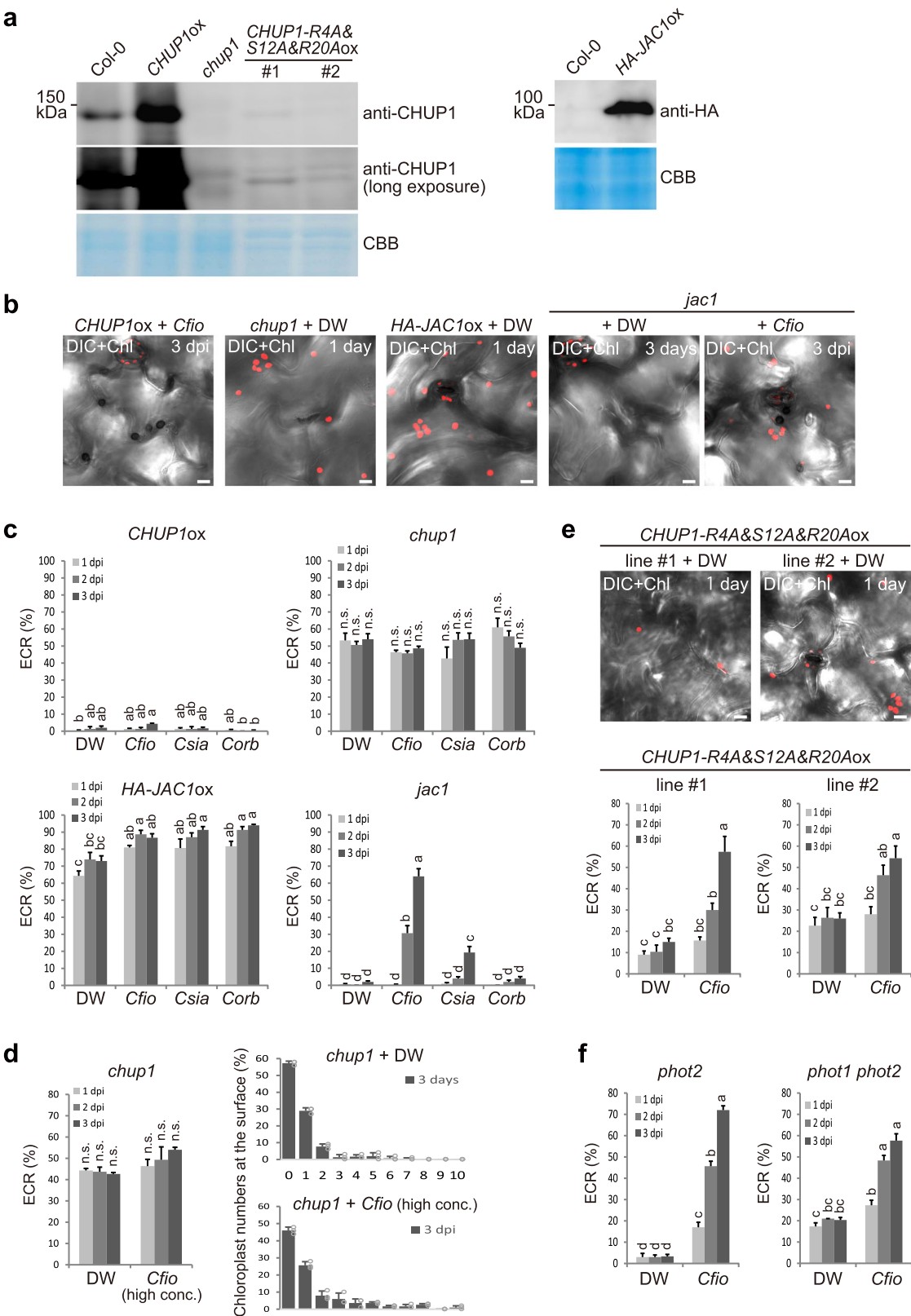

showed a greater decrease compared to that in *pen2*, suggesting the involvement of the epidermal chloroplast-localized proteins GSH1, EDS5, and CAS in the expression of these genes (Fig. 7b). It was reported that *PR1* and *FRK1* expression is regulated by the SA signaling pathway[55,56]. Therefore, our data on *PR1* and *FRK1*

expression are consistent with the fact that EDS5 and CAS play a role in SA accumulation[17,52,53,57]. We also found that a mutation of *SID2* involved in SA biosynthesis increased *Colletotrichum* entry into the epidermis in the *pen2* background (Supplementary Fig. 10). Interestingly, Rekhter et al. previously provided images

**Fig. 3 CHUP1 and JAC1 regulate the ECR in a phototropin-independent manner. a** Levels of CHUP1 and HA-JAC1 proteins in the indicated *Arabidopsis*. Protein extracts were analyzed by immunoblot with anti-CHUP1 and anti-HA antibodies. CBB staining was used as a loading control. **b, e** Effects of CHUP1 and JAC1 on *Cfio*-induced ECR. *Cfio* was inoculated onto cotyledons of *CHUP1*ox and *jac1*, and the surface of the epidermis was observed. DW was spotted onto *chup1*, *CHUP1-R4A&S12A&R20A*ox lines, *HA-JAC1*ox, and *jac1* as a control. The epidermal chloroplasts were visualized based on chlorophyll autofluorescence. The DIC images were captured by confocal microscopy at the indicated time point. Scale bar, 10 μm. **c–e** Quantification of ECR in *CHUP1*ox, *chup1*, *HA-JAC1*ox, *jac1*, and *CHUP1-R4A&S12A&R20A*ox plants. *Cfio*, *Csia*, and *Corb* ($5 \times 10^5$ conidia/mL) were inoculated onto cotyledons of the indicated *Arabidopsis* and examined at 1, 2, and 3 dpi (**c, e**). For more fungal inoculum on the *chup1* mutant, *Cfio* ($1.5 \times 10^6$ conidia/mL) was inoculated (**d**). A total of 100 cells in contact with the melanized appressorium were observed. As a control, DW was spotted. **f** *Cfio*-induced ECR in *phot2* and *phot1 phot2* mutants. As a control, DW was spotted. The means and SE were calculated from three independent plants. Means not sharing the same letter are significantly different ($P < 0.05$, two-way ANOVA with Tukey's HSD test). n.s. not significant.

showing preferential localization of ICS1/SID2 in small epidermal chloroplasts compared to large mesophyll chloroplasts, although they did not refer to this point in the text[53].

Notably, the double and triple mutants tested above retained sufficient preinvasive defense against *Corb* (Figs. 4a, 7a). To clarify the involvement of these components, we systematically generated multiple mutant plants. Remarkably, the preinvasive NHR against *Corb* was abolished in correlation with the number of mutations; the invasive hyphae were clearly observed (Fig. 7c left, 7d upper). Along with the higher entry rates of *Cfio* and *Csia* in multiple mutant plants (Supplementary Fig. 11), these results suggest that the epidermal chloroplast-related factors ECR, GSH1, EDS5, and CAS are common NHR determinants that contribute to preinvasive immunity against *Colletotrichum* fungi in *Arabidopsis*. On the other hand, many plants with multiple mutations showed enhanced susceptibility to *Chig*, suggesting that these factors also contribute to basal resistance against adapted *Colletotrichum* fungus (Supplementary Fig. 12).

Our data also suggested that the ECR had somewhat additive effects with GSH1, EDS5, and CAS on NHR against *Corb* (Fig. 7d upper). For this reason, we speculate that epidermal chloroplasts of *Arabidopsis* might carry other components engaged in additional immune pathway(s) that are effective against highly incompatible *Corb*. It is also possible that GSH1, EDS5, and CAS can partially function in preinvasive NHR against *Corb* in the absence of the ECR. Nevertheless, we still expected the presence of functional redundancy with GSH1, EDS5, and CAS with the ECR: GSH1, EDS5 and CAS partially have overlapping functions with the ECR for preinvasive NHR, because we here revealed that (i) the ECR regulates the intracellular localization of GSH1, EDS5, and CAS, and (ii) both the ECR and these immune components are required for preinvasive NHR against *Corb*. The ECR might exhibit functional redundancy with GSH1, EDS5, and CAS on NHR against *Cfio* and *Csia*; however, their entry rates on many multiple mutants of *Arabidopsis* reached 80% or higher (Supplementary Fig. 11). To clarify this point, we searched for additional *Colletotrichum* fungi and found the *Csia* strain COC4, which was isolated from the hau tree and had low invasion ability into *Arabidopsis* mutants. Similar to *Corb*, this *Csia* (low-invasive strain) induced lower levels of the ECR in Col-0 (Supplementary Fig. 13), but showed a higher entry rate on multiple mutants than *Corb* (Fig. 7c, d). Importantly, ECR-deficient mutation exhibited redundancy with *gsh1*, *eds5*, and *cas* mutations in the context of preinvasive NHR against the low-invasive *Csia* strain (e.g., *edr1 pen2 gsh1 eds5 cas* vs. *edr1 pen2 gsh1 eds5 cas chup1*). These results indicate a functional link between the ECR and these immune components in NHR (Fig. 7d lower).

**The ECR strongly occurs and contributes to NHR against specific nonadapted *Colletotrichum* fungus in wild-type *Arabidopsis*.** The ECR was strongly induced and contributed to NHR in the *pen2* mutant (Figs. 1c and 4), implying that the ECR ensures preinvasive immunity in NHR when higher-layer

preinvasive defenses such as PEN2-related pathway are ineffective. To elucidate the role of the ECR in *Arabidopsis* under natural condition, we searched for nonadapted *Colletotrichum* fungi with higher invasion ability than *Cfio*. We found that *C. nymphaeae* PL1-1-b (*Cnym*), isolated from a Japanese flowering cherry, developed small visible lesions on wild-type *Arabidopsis* Col-0 compared to *Cfio*, but its progression was tightly restricted (Fig. 8a). Thus, *Cnym* is not adapted to *Arabidopsis* in comparison to the pathogenicity of adapted *Chig*, while *Cnym* could partly invade epidermal cells (Fig. 8a, b). Consistent with the degree of lesion formation in Col-0, the entry rate of *Cnym* was higher than that of *Cfio* (Figs. 1a, e and 8a, c). Importantly, *Cnym* induced the ECR at a high frequency in Col-0, indicating that the ECR strongly occurs against the nonadapted fungus *Cnym* in wild-type *Arabidopsis* (Fig. 8d). Defense-related genes were also induced in Col-0 by *Cnym* inoculation, which contrasts with the results observed after *Cfio* inoculation (Supplementary Figs. 2 and 14). *Cnym*-induced ECR was moderately elevated in the *pen2* mutant with increased fungal entry (Fig. 8c, d), suggesting that PEN2-related immunity was still partially working against *Cnym*. Importantly, *CHUP1*ox and *chup1* single mutants exhibited a significant decrease and decreasing trend, respectively, in preinvasive defense against *Cnym* compared to Col-0; meanwhile, the ECR-normal *phot2* mutant showed resistance comparable to that of Col-0 (Fig. 8c, d). Collectively, these results strongly suggest that the ECR is highly activated and contributes to preinvasive NHR, at least against nonadapted fungus *Cnym* in wild-type *Arabidopsis* plants.

**The ECR and immune components GSH1, EDS5, and CAS are involved in NHR against *M. oryzae*.** To assess the function of the ECR in plant immunity against appressorium-mediated entry of other fungi, we inoculated *A. thaliana* with nonadapted *M. oryzae* (Fig. 9). The ECR also occurred in response to the entry trial of *M. oryzae*, and this response increased in the *pen2* mutant, suggesting a similar epidermal response to that observed for *Colletotrichum* fungi (Fig. 9a, b). As has been previously reported[6], *M. oryzae* invaded the *pen2* mutant to a slight degree, whereas the *CHUP1*ox plant showed comparable resistance to Col-0 (Fig. 9c). Moreover, both *pen2 CHUP1*ox and *pen2 chup1* plants exhibited a higher reduction in preinvasive defense compared to *pen2*; the pathogen sufficiently developed invasive hyphae through melanized appressorium but was confined in the dead epidermal pavement cell, both of which were clearly visualized by TB staining (Fig. 9b–d). In contrast, the apoplastic bacterial pathogen *Pseudomonas syringae*, which cannot penetrate the epidermis but enters through natural openings (e.g., stomata) and multiplies in the apoplastic space, did not induce the ECR (Fig. 9e, Supplementary Fig. 15). These findings further support the idea that the trigger for the ECR is specific to the fungal penetration stage. Consistent with this, the susceptibility to *P. syringae* was not affected in ECR-defective plants (Fig. 9f). We also investigated the effects of *gsh1*, *eds5*, and *cas* mutations on preinvasive NHR

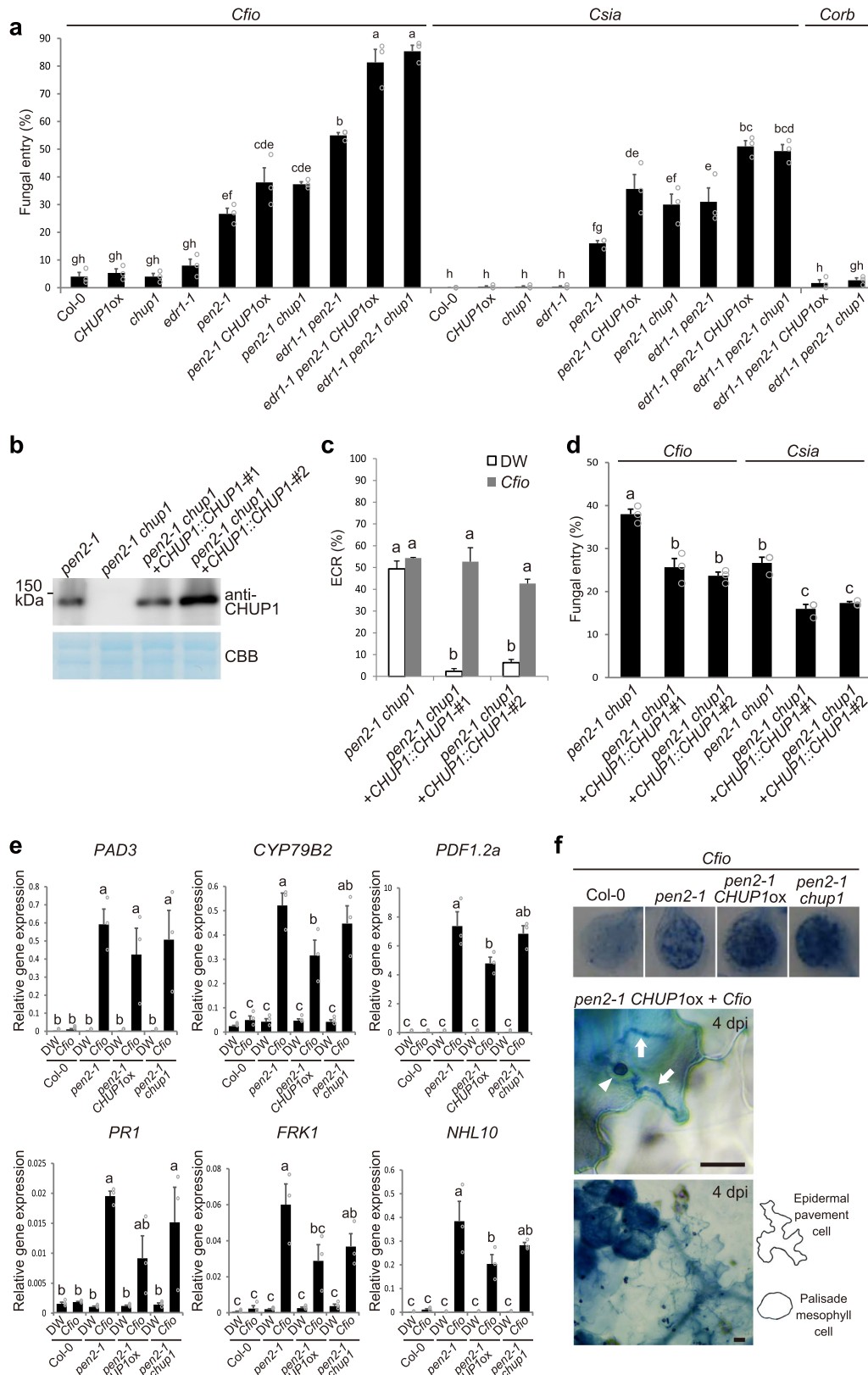

against *M. oryzae* and found that *M. oryzae* showed a greater increase in entry into all the double mutants with *pen2* mutation (Fig. 9c). These results suggest that *A. thaliana* deploys similar immune responses via epidermal chloroplasts to prevent the appressorium-mediated entry of *Colletotrichum* and *Magnaporthe* fungi.

**Epidermal chloroplasts support the intracellular movement of other organelles in ECR-activated cells.** We frequently observed that several chloroplasts clustered around the presumed nucleus at the surface of ECR-activated cells (Figs. 1b, 3b, 5 and 9a). In the mesophyll and epidermal pavement cells of *Arabidopsis*, the nuclei move together with chloroplasts in a phototropin-

**Fig. 4 The ECR contributes to *Arabidopsis* NHR against *Colletotrichum* fungi. a** Fungal entry rate into the epidermis of ECR-defective *Arabidopsis*. A total of 100 melanized appressoria were examined at 4 dpi. **b** CHUP1 protein levels in the indicated plants. Protein extracts were analyzed by immunoblot with anti-CHUP1 antibody. CBB staining was used as a loading control. **c** ECR of *pen2-1 chup1* and CHUP1-complementation lines. A total of 100 cells in contact with the melanized appressorium were examined at 2 dpi. DW was used as a control. **d** Entry rate of *Cfio* and *Csia* into the epidermis of *pen2-1 chup1* and CHUP1-complementation lines at 4 dpi. **e** Induced gene expression of *PAD3*, *CYP79B2*, *PDF1.2a*, *PR1*, *FRK1*, and *NHL10* by inoculation of *Cfio*. *Cfio* was inoculated onto cotyledons of the indicated *Arabidopsis*. Gene expression was assayed at 24 hpi by RT-qPCR. DW was used as a control. **f** Epidermal and mesophyll cell death caused by appressorium-mediated entry of *Cfio* at 4 dpi. The inoculated plants were subjected to TB staining and observed macroscopically (upper) and microscopically using x40 (middle) and x10 (lower) objective lenses. The arrowhead and arrows indicate melanized appressorium and invasive hyphae, respectively. Scale bar, 20 µm. For all quantification analyses except RT-qPCR, the means and SE were calculated from three independent plants. The means and SE of RT-qPCR results were calculated from three independent experiments. Means not sharing the same letter are significantly different (*P* < 0.05, two-way ANOVA with Tukey's HSD test).

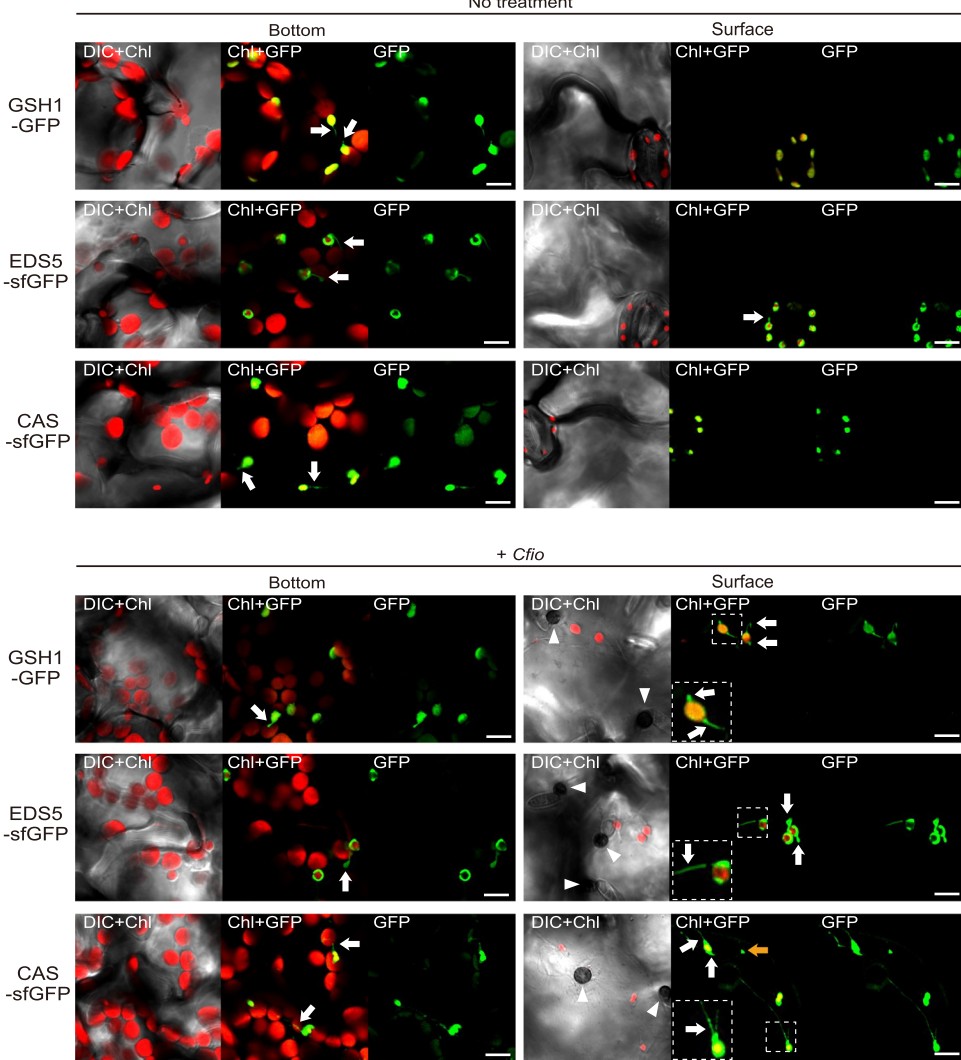

**Fig. 5 Epidermal chloroplasts preferentially position immune components GSH1, EDS5, and CAS.** Localization patterns of GSH1, EDS5, and CAS proteins in *Arabidopsis*. The transgenic *gsh1* mutant expressing GSH1-GFP[46] under its own promoter and Col-0 plants expressing EDS5-sfGFP and CAS-sfGFP under the CaMV *35S* promoter were incubated with or without *Cfio* and observed at 2 dpi. Chloroplasts were visualized using chlorophyll autofluorescence. The lower left dashed-line square regions show a magnified image of the dashed box. The white arrowheads and arrows indicate melanized appressoria and stromules, respectively. The orange arrow indicates CAS localized in the epidermal plastid without chlorophyll. Scale bar, 10 µm. Similar results were obtained from three independent experiments.

dependent manner[54,58]. Moreover, perinuclear clustering of chloroplasts, stromule induction, and stromule-mediated chloroplast-to-nucleus ROS signaling in response to viral effectors have been reported in *N. benthamiana*[18,59,60]. To confirm whether nuclei showed an ECR-like response after fungal attack, we simultaneously visualized the nuclei and chloroplasts of *Cfio*-inoculated *Arabidopsis* (Fig. 10). DAPI-stained nuclei with several chloroplasts emerged at the epidermal surface in response to *Cfio*, whereas no nuclei were observed at the epidermal surface without fungal inoculation (Fig. 10a). We also observed CFP-labeled endoplasmic reticulum (ER) around the presumed nuclei with a cluster of chloroplasts in the ECR-activated cells (Fig. 10b). These

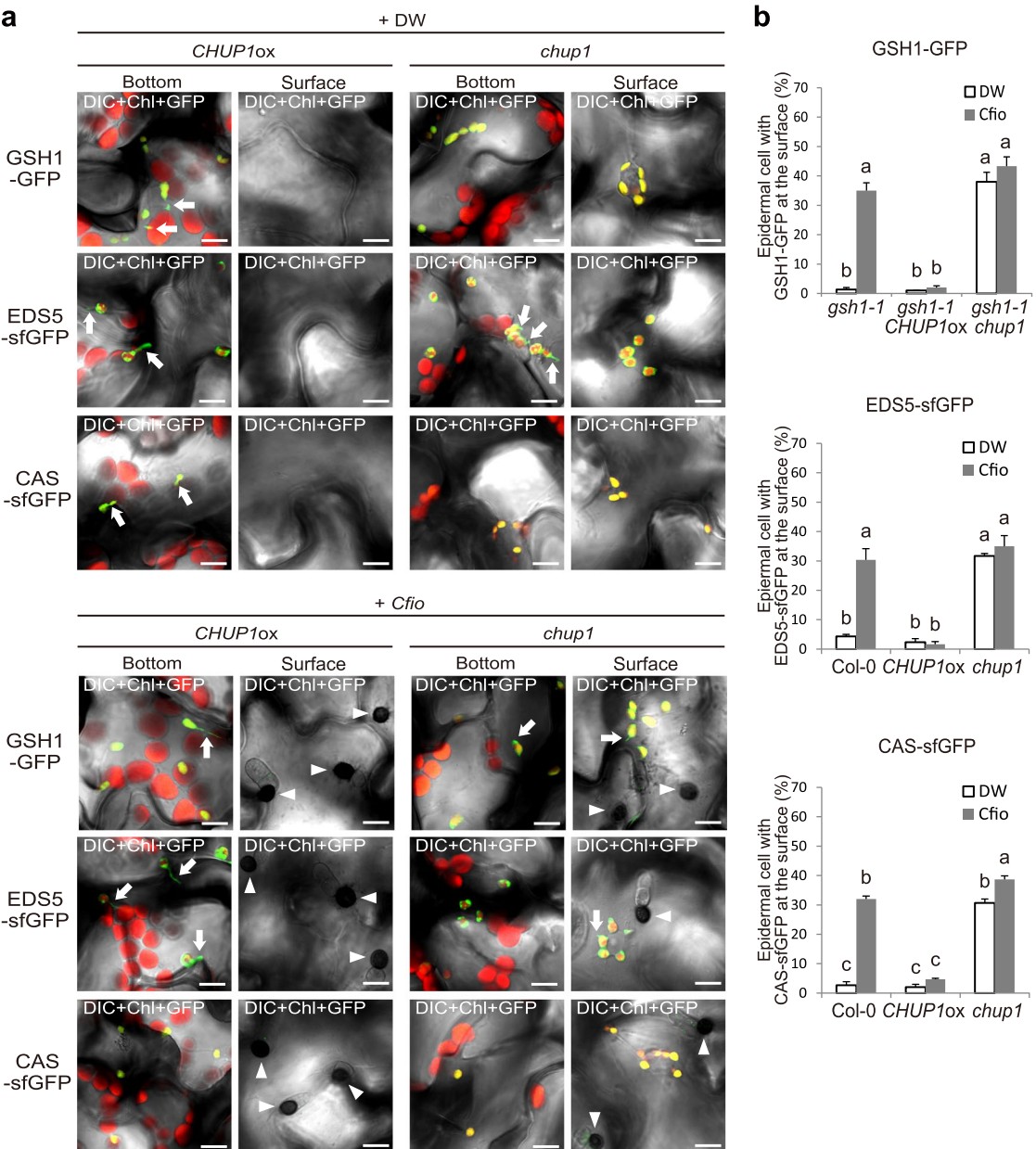

**Fig. 6 Localization pattern of GSH1, EDS5, and CAS tightly linked to epidermal chloroplasts and the ECR. a** Localization pattern of GSH1, EDS5, and CAS proteins in *CHUP1*ox and *chup1* plants. The transgenic *gsh1 CHUP1*ox and *gsh1 chup1* plants expressing GSH1-GFP under its own promoter, and *CHUP1*ox and *chup1* plants expressing EDS5-sfGFP and CAS-sfGFP under the CaMV *35S* promoter were incubated with *Cfio* and observed at 2 dpi. DW was used as a control. The chloroplasts were visualized based on chlorophyll autofluorescence. The white arrowheads and arrows indicate melanized appressoria and stromules, respectively. Scale bar, 10 μm. **b** Quantification of epidermal cells with surface fluorescence of GFP/sfGFP-labeled immune components. *Cfio*-inoculated cotyledons of the indicated *Arabidopsis* were analyzed at 2 dpi. A total of 100 cells in contact with the melanized appressorium were observed. As a control, DW was spotted. The means and SE were calculated from three independent plants. Means not sharing the same letter are significantly different (*P* < 0.05, two-way ANOVA with Tukey's HSD test).

results indicate that epidermal chloroplasts and nuclei surrounded by the ER dynamically respond to the fungal entry trial.

It has been reported that perinuclear chloroplast movement is required to generate the motive force of nuclear movement in *Arabidopsis* epidermis[54]. Consistent with this, the nuclei of the *CHUP1*ox and *chup1* plants that exhibited ineffective ECR showed a similar behavior to that of perinuclear chloroplasts (Fig. 10c, d). Since CHUP1 localizes only to the epidermal and mesophyll chloroplast envelopes and not to the nuclear envelope[54], it is suggested that CHUP1-dependent ECR repression results in the inhibition of nuclear movement to the

epidermal surface with *Cfio* inoculation. These results indicate that epidermal chloroplasts also function as critical factors supporting the movement of other organelles via the ECR during the antifungal defense response.

## Discussion

Typical large mesophyll chloroplasts are important plant organelles that are responsible for photosynthesis. Guard cell chloroplasts have recently been reported to regulate stomatal movement in response to $CO_2$ and light[23]. However, although small chloroplasts in the epidermal pavement cells of *A. thaliana*

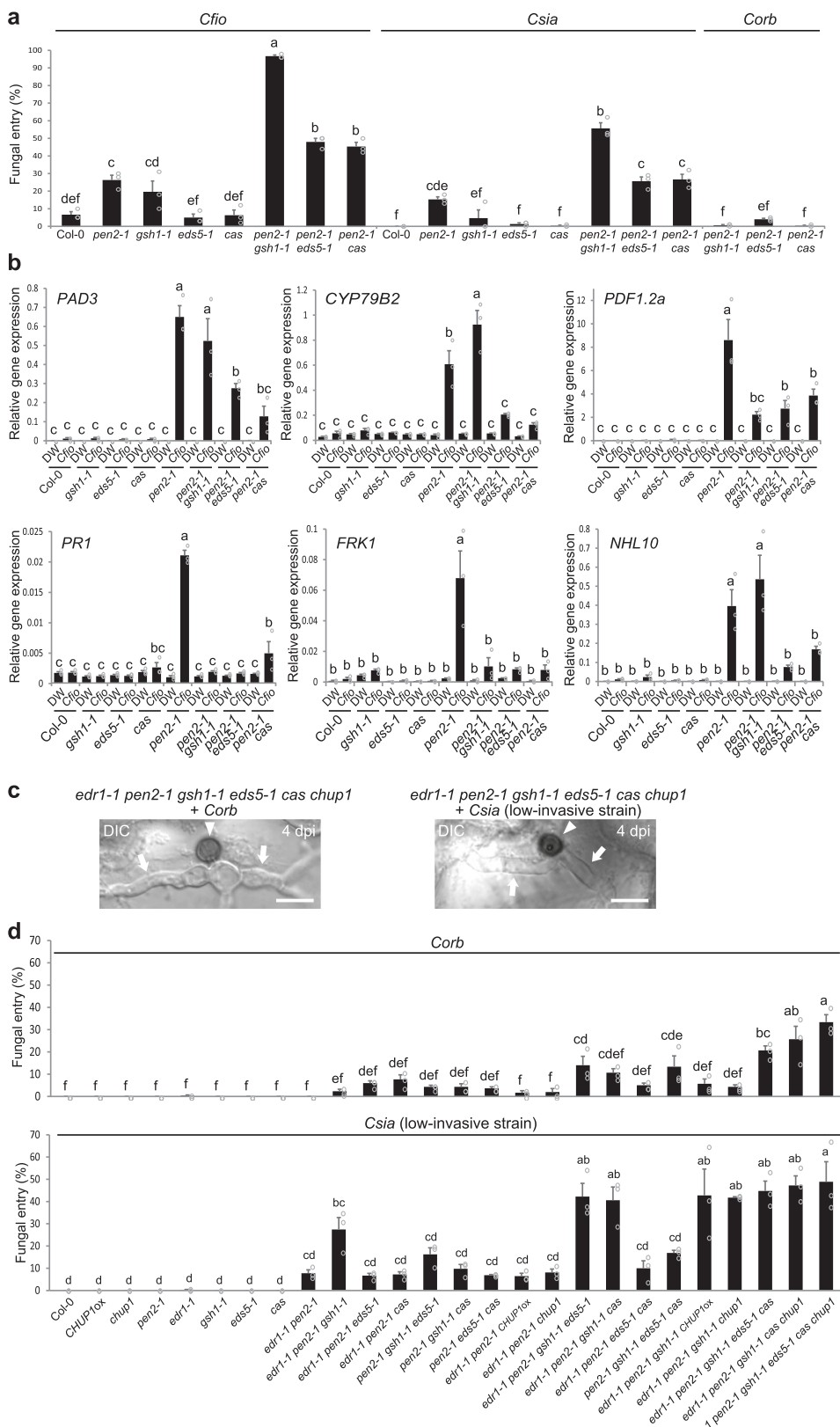

have been recognized[24,25], their functional significance remains unknown.

In this study, we discovered a dynamic response of epidermal chloroplasts in response to fungal pathogens (Supplementary Fig. 16). More precisely, the response occurred against appressorium-mediated entry trials of nonadapted *Colletotrichum*

and *Magnaporthe* fungi. We designated this phenomenon as the ECR. The ECR was activated in the lower layer of PEN2-related immunity (Fig. 1c, Supplementary Fig. 13); therefore, it may not be the first layer of preinvasive NHR. We concluded that the ECR is programmed to be activated when higher-layer preinvasive NHR, such as the PEN2-related pathway, is not effective; this is

**Fig. 7 Epidermal chloroplast-localized immune components are critical for preinvasive NHR against *Colletotrichum* fungi. a** Fungal entry rate into *Arabidopsis* epidermis. *Cfio*, *Csia*, and *Corb* were inoculated onto cotyledons of the indicated *Arabidopsis*. **b** Induced gene expression of *PAD3*, *CYP79B2*, *PDF1.2a*, *PR1*, *FRK1*, and *NHL10* by inoculation of *Cfio*. *Cfio* was inoculated onto cotyledons of the indicated *Arabidopsis*, and gene expression was assayed at 24 hpi by RT-qPCR. DW was used as a control. **c**, **d** Invasions of *Corb* and *Csia* (low-invasive strain) into *Arabidopsis* epidermis at 4 dpi. *Corb* and *Csia* (low-invasive strain) were inoculated onto cotyledons of the indicated *Arabidopsis* with multiple mutations. The arrowheads and arrows indicate melanized appressoria and invasive hyphae, respectively. Scale bar, 10 μm. For quantification of fungal entry of *Cfio*, *Csia*, and *Corb*, 100 melanized appressoria were investigated at 4 dpi. For quantification of fungal entry of *Csia* (low-invasive strain), 300 melanized appressoria were investigated at 4 dpi. For all quantification analyses except RT-qPCR, the means and SE were calculated from three independent plants. The means and SE of RT-qPCR results were calculated from three independent experiments. Means not sharing the same letter are significantly different ($P < 0.05$, one-way (**d**) or two-way (**a**, **b**) ANOVA with Tukey's HSD test).

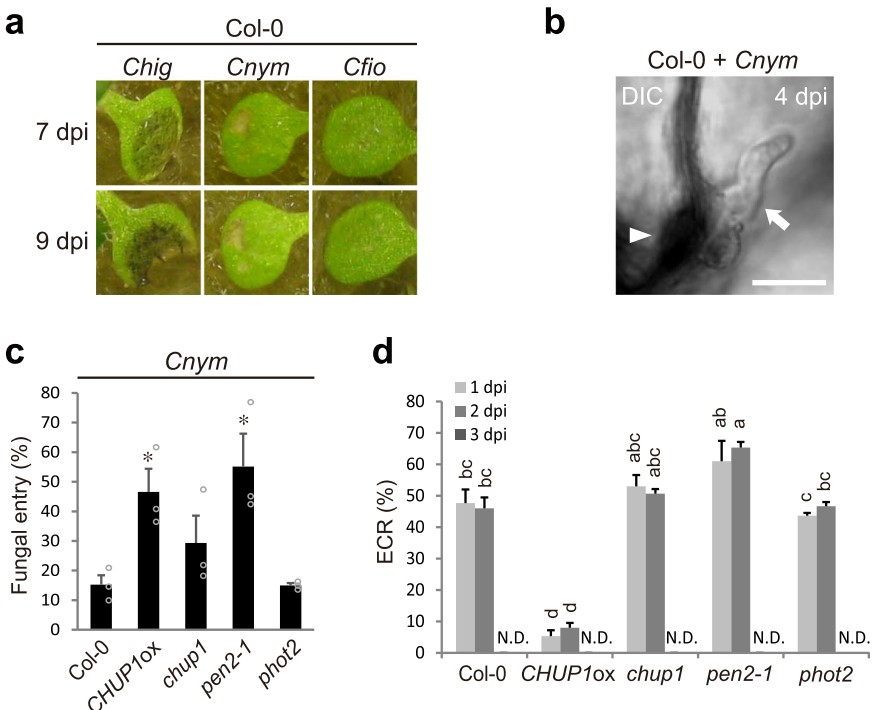

**Fig. 8 The ECR strongly occurs and contributes to NHR against specific nonadapted *Colletotrichum* fungus in wild-type *Arabidopsis*. a** Pathogenicity of *Colletotrichum* fungi on wild-type *Arabidopsis* (Col-0). A conidial suspension of adapted *Chig* and nonadapted *Cnym* and *Cfio* was inoculated onto cotyledons of Col-0 and incubated for 7 and 9 days. **b**, **c** *Cnym* invasion into epidermis of Col-0 at 4 dpi. The arrowhead and arrow indicate melanized appressorium and invasive hypha, respectively. Scale bar, 10 μm. For quantification of fungal entry of *Cnym*, 300 melanized appressoria were investigated at 4 dpi. **d** Quantification of ECR in Col-0, *CHUP1*ox, *chup1*, *pen2-1*, and *phot2* plants after inoculation of *Cnym* at 1, 2, and 3 dpi. A total of 100 cells in contact with the melanized appressorium were observed. For all quantification analyses, the means and SE were calculated from three independent plants. Means not sharing the same letter are significantly different ($P < 0.05$, two-way ANOVA with Tukey's HSD test). The asterisks indicate significant difference (*$P < 0.05$, one-way ANOVA with Dunnett's test).

because a specific nonadapted fungus, which partly overcomes higher-layer preinvasive defense(s), induces the greatest ECR in the wild-type *Arabidopsis* background (Fig. 8).

The trigger for the ECR remains unknown. We originally considered PAMPs or DAMPs as candidates; indeed, almost all *Cfio*-induced genes in *pen2* plants (Supplementary Fig. 2) are PAMP- or DAMP-inducible[5,34,61–63]. However, the ECR did not depend on the PAMP and DAMP signaling kinases BAK1, BIK1, and PBL1, nor on the DAMP receptors PEPR1 and PEPR2 (Supplementary Fig. 3). Instead, we found that fungal peg formation and secretion activity were strongly correlated with the ECR, suggesting penetration stage-specific pathogen recognition by *A. thaliana* (Fig. 2). *Colletotrichum* effectors are secreted from the penetration pore formed at the base of the appressorium[38,64]. A specific effector conserved at least in *Colletotrichum* and *Magnaporthe* is a candidate trigger for the ECR, although the recognition of pathogen effectors usually induces a hypersensitive response. However, if the ECR is a more universal process against

many types of fungal pathogens, plant cell wall damage caused by plant cell wall-degrading enzymes (PCWDEs), which are commonly secreted by fungi, or such PCWDEs themselves might trigger the ECR. Therefore, it is possible that unknown PAMP or DAMP signaling independent of BAK1, BIK1, PBL1, PEPR1, and PEPR2 is involved in this process.

Five nonadapted *Colletotrichum* strains displayed different abilities to trigger the ECR; *Cnym* induced the strongest ECR, and *Cfio* induced the ECR more strongly than *Csia* (Figs. 1c and 8d). Meanwhile, *Csia* (low-invasive strain) and *Corb* almost never induced the ECR, in Col-0 (Fig. 1c, Supplementary Fig. 13). All these fungi induced a rather strong ECR in *pen2* (Figs. 1c and 8d, Supplementary Fig. 13). This might reflect their differential ability to overcome the higher-layer preinvasive defenses of *Arabidopsis*. *Cnym* partly invaded Col-0 and *Cfio* invaded Col-0 to a slight degree but invaded *pen2* more, *Csia* could invade *pen2* but not Col-0, and *Csia* (low-invasive strain) and *Corb* could not invade *pen2* (Figs. 1e, 7d and 8c). Remarkably, *Cfio* induced the

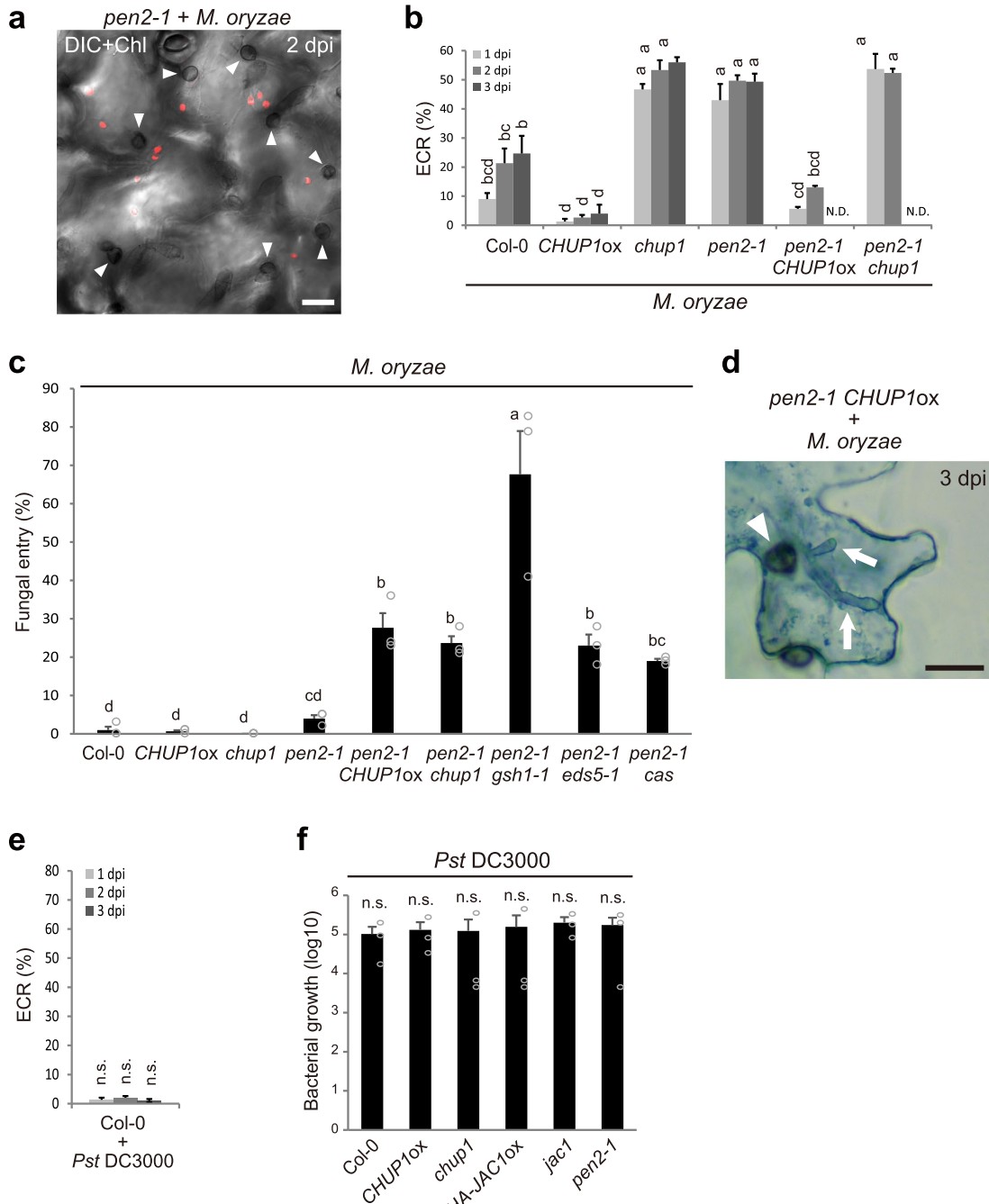

**Fig. 9 The ECR contributes to preinvasive NHR against _M. oryzae_. a** ECR after inoculation of _M. oryzae_. _M. oryzae_ was inoculated onto cotyledons of _pen2-1_ plants, and the ECR was investigated at 2 dpi. The chloroplasts were visualized based on chlorophyll autofluorescence. **b** Quantification of ECR against _M. oryzae_. ECR was investigated at 1, 2, and 3 dpi. A total of 100 cells in contact with the melanized appressorium were observed. N.D.: not determined due to damages of epidermal cell by fungal invasion. **c** Entry rate of _M. oryzae_ into epidermis at 3 dpi. A total of 100 melanized appressoria were investigated. **d** Epidermal cell death caused by appressorium-mediated entry of _M. oryzae_ at 3 dpi. The inoculated plants were subjected to TB staining and observed microscopically using x40 objective lens. The arrowheads and arrows indicate melanized appressoria and invasive hyphae, respectively. Scale bar, 20 μm. The means and SE were calculated from three independent plants. Means not sharing the same letter are significantly different ($P < 0.05$, one-way (**c**) or two-way (**b**) ANOVA with Tukey's HSD test). **e** Quantification of ECR against _Pseudomonas syringae_ pv. _tomato_ (_Pst_) DC3000. Cotyledons of Col-0 were drop-inoculated with _Pst_ DC3000, and ECR was investigated at 1, 2, and 3 dpi. A total of 100 epidermal pavement cells were observed. The means and SE were calculated from three independent plants. **f** Growth of _Pst_ DC3000 in _Arabidopsis_ cotyledons. Cotyledons of indicated _Arabidopsis_ were drop-inoculated with _Pst_ DC3000, and incubated for 4 days. The number of bacteria in eight cotyledons obtained from four independent plants was plotted on a log10 scale. The means and SE were calculated from three independent experiments. n.s. not significant (*$P < 0.05$, one-way (**f**) or two-way (**e**) ANOVA with Tukey's HSD test).

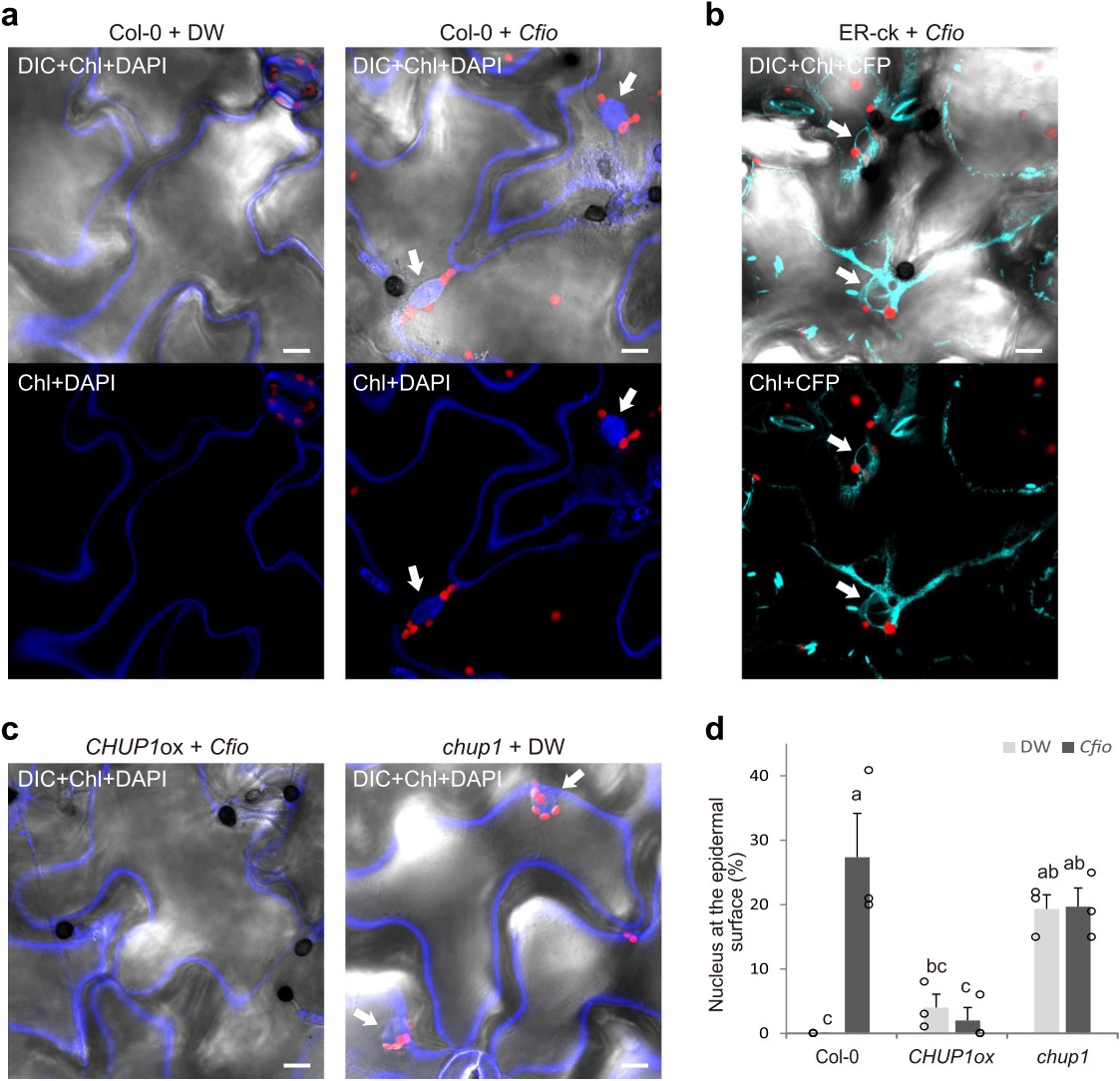

**Fig. 10 The nucleus dynamically responds to fungal entry trial in an ECR-dependent manner. a** Simultaneous observation of chloroplasts and nuclei in ECR-activated epidermis. The surface of *Cfio*-inoculated cotyledons of Col-0 was examined at 3 dpi. The chloroplasts and nuclei were visualized by chlorophyll autofluorescence and DAPI staining, respectively. DW was used as a control. The arrows indicate nuclei. **b** Simultaneous observation of chloroplasts and ER around nuclei in ECR-activated cells. The surface of *Cfio*-inoculated cotyledons of the transgenic plant line ER-ck expressing ER-CFP was observed at 3 dpi. The arrows indicate perinuclear ER. **c** Simultaneous observation of chloroplasts and nuclei in ECR-defective plants at 3 dpi. *Cfio*-inoculated cotyledons of *CHUP1*ox plants were investigated. DW was spotted onto *chup1* mutant as a control. The arrows indicate nuclei. **d** The ratio of epidermal cells with surface nuclei was investigated at 3 dpi. A total of 100 cells in contact with the melanized appressorium were observed. Scale bar, 10 μm. The means and SE were calculated from three independent plants. Means not sharing the same letter are significantly different ($P < 0.05$, two-way ANOVA with Tukey's HSD test).

expression of defense-related genes in *pen2* but not in Col-0 at 24 h post-inoculation (hpi) (Supplementary Fig. 2). Most of these genes were also not induced in Col-0 at 72 hpi of *Cfio*, while *Cnym* moderately induced these genes at 72 hpi (Supplementary Figs. 2 and 14). These results suggest that the threshold at which the ECR occurs in each epidermal cell is lower than that for the induction of defense-related genes. However, the population of ECR-activated epidermal cells gradually increased throughout the pathogen-inoculated leaf during incubation with the pathogen, which might activate the expression of defense-related genes (Fig. 1b, c, Supplementary Fig. 2). Thus, the ECR could be a good indicator to evaluate whether fungal pathogens overcome higher-layer preinvasive defense(s) even in a slight degree.

Importantly, we demonstrated that the ECR is an actual pre-invasive NHR determinant against fungal pathogens. The ECR was under the control of CHUP1 and JAC1, which are regulators of chloroplast photorelocation in mesophyll cells (Supplementary Fig. 16). The excess or depletion of CHUP1 proteins resulted in impairments of the ECR (Figs. 3, 8d and 9b). Furthermore, these plants showed decreased preinvasive NHR against the entry of nonadapted fungi in the *pen2* background (Figs. 4 and 9c). CHUP1 is a critical factor for the ECR and avoidance and accumulation responses in the mesophyll photo-relocation system, whereas the ECR and avoidance response of *jac1* were near-normal and normal, respectively (Fig. 3)[40,41]. This indicates that the ECR has a regulatory mechanism similar to the mesophyll chloroplast avoidance response. Since only the photorelocation movement of large mesophyll chloroplasts has been well-studied[39], our findings revealed that another type of chloroplast in the epidermis also moves under fungal attack. Although the signal input system is distinct, light and pathogen recognition, respectively, the sharing of downstream components

implies the versatility of the optical response system for plant immunity.

We also aimed to explore the mechanism underlying the involvement of the ECR in antifungal NHR. We showed that the immune components GSH1, EDS5, ICS1/SID2, and CAS contributed to preinvasive NHR (Figs. 7 and 9c, Supplementary Figs. 10 and 11). Importantly, these components were preferentially localized to epidermal chloroplasts, regardless of the fungal entry trials (Fig. 5, Supplementary Fig. 8, Supplementary Movies 2–7)[53]. Although a previous report has shown that CAS-GFP expressed constitutively also localized to mesophyll chloroplasts, the levels of CAS proteins showed large variation[50]. Therefore, the expression levels of CAS-sfGFP in our experiment might have been lower than those in the previous study. Alternatively, sfGFP, with its more improved folding efficiency and higher fluorescence intensity in planta than other GFP variants[65], might make a difference in the localization pattern of CAS protein in our experiment. Remarkably, *gsh1*, *eds5*, *sid2*, and *cas* mutations exhibited an increased fungal entry rate, especially in the absence of PEN2-related defense (Fig. 7a, Supplementary Fig. 10). Plants with multiple mutations in these components showed more severe breakdown of preinvasive NHR (Fig. 7d, Supplementary Fig. 11). More importantly, we showed that the ECR-deficient mutation acted redundantly with *gsh1*, *eds5*, and *cas* mutations in the preinvasive NHR against the low-invasive *Csia* strain (Fig. 7d lower). These results suggest that the movement of these immune components via the ECR is a key factor for epidermis-specific antifungal NHR (Supplementary Fig. 16). In contrast, rice SNARE protein OsVAMP714 localizes to both mesophyll chloroplasts and the epidermal vacuolar membrane, and mesophyll localization is more important for the resistance against adapted *M. oryzae*[66]. Thus, the functional differences between epidermal and mesophyll chloroplasts in plant immunity remain to be elucidated.

The involvement of CHUP1-mediated chloroplast movement to the host–pathogen interface in immunity to the oomycete pathogen *Phytophthora infestans* was proposed in *N. benthamiana*[67]. Thus, partially similar immune mechanisms involving chloroplast movement may exist in *A. thaliana* and *N. benthamiana*. In contrast, we occasionally observed epidermal chloroplasts near the appressoria in ECR-activated cells, and the majority of the population in the surface chloroplasts was scattered. This is slightly different from the observations in *N. benthamiana*[67]. Moreover, unlike *Arabidopsis* ECR (Supplementary Fig. 3), *P. infestans*-induced stromule formation in *N. benthamiana* depended on BAK1-mediated signaling[67], implying the presence of a different pathogen recognition system from *Arabidopsis* ECR. In addition, we revealed that epidermal chloroplast-associated nuclei surrounded by ER also showed an ECR-like response in an ECR-dependent manner (Fig. 10). It is known that many plant organelles, including the nucleus, move to the site of attempted penetration by fungal pathogens, including *Colletotrichum* fungi[68–71]. Although we occasionally observed nuclear translocation associated with epidermal chloroplasts near the fungal appressoria in ECR-activated cells, ECR-dependent intracellular nuclear movement may also be related to antifungal NHR. Meanwhile, there is a possibility that the ECR supports the movement of other epidermal immune-related organelles, in addition to the nuclei and ER. The link between chloroplasts and EDR1 has not yet been reported. However, EDR1 localizes to the nucleus, ER, cytoplasm, and trans-Golgi network/early endosomes[11,72,73]; therefore, EDR1 might be associated with the ECR through the movement of other organelles.

The immune components GSH1, EDS5, and CAS expanded their location from the epidermal chloroplasts to stromules (Figs. 5 and 6, Supplementary Fig. 8, Supplementary Movies 2–7).

Since stromules are considered facilitators of ROS signaling and immune protein transport from chloroplasts to the nucleus in *N. benthamiana*, the same is probably true for *Arabidopsis*[18]. Therefore, our data indicate the potential link between immune-related *Arabidopsis* proteins and stromules. Stromules have also been reported to be associated with other organelles and sub-cellular structures[74]. Further studies are required to investigate how stromules contribute to ECR-mediated immunity by transporting many components. Intriguingly, the *Arabidopsis chup1* mutant exhibited constitutive stromule formation and an increased hypersensitive response cell death in response to bacterial effectors as a defense response[18]. Thus, the decreased preinvasive NHR against *Colletotrichum* fungi caused by *chup1* mutation with constitutive stromule formation further supported the importance of the ECR itself for antifungal immunity (Figs. 4 and 8c, d).

Collectively, our findings shed light on the dynamic responses of atypical small chloroplasts in the plant epidermis. We propose that *Arabidopsis* epidermal cells deploy an ECR-centered dynamic network, wherein the intracellular traffic of small signaling molecules and immune-related proteins might easily occur via the ECR, and possibly stromules and other organelle movements, to support intracellular signaling and hence antifungal NHR. The plant epidermal cell is the primary stronghold that combats environmental pathogens; therefore, another type of chloroplast in the epidermis may have evolved as a defense-related organelle. Future studies on this system in the ECR-activated epidermis will expand our understanding of plant defense measures at the point of contact with fungal pathogens.

## Methods

**Fungal strains and media**. *Colletotrichum higginsianum* Abr1-5 (MAFF305635), *Colletotrichum nymphaeae* PL1-1-b (MAFF240037), *Colletotrichum fioriniae* CC1 (MAFF306550), *Colletotrichum siamense* MAF1 (MAFF243010), COC4 (MAFF243696), and *Magnaporthe oryzae* Ai79-142 (MAFF101520) were obtained from the Ministry of Agriculture, Forestry and Fisheries GenBank, Japan. The wild-type *Colletotrichum orbiculare* (syn. *C. lagenarium*) strain 104-T (MAFF240422) was stored at the Laboratory of Plant Pathology, Kyoto University. *C. orbiculare* transgenic strains, 104-T expressing *CoDN3::CoDN3-mCherry*, and Δ*sec22* expressing *CoDN3::CoDN3-mCherry*[38], and the disruption mutants, *cst1*[30], *atg26*[32], and *icl1*[33], have been described previously. Cultures of all fungal isolates of *Colletotrichum* were maintained on 2.5% or 3.9% (w/v) PDA medium (Nissui Pharma or BD Difco) at 24 °C in the dark. *M. oryzae* culture was maintained on oatmeal agar medium at 24 °C in the dark. For conidiation, *C. nymphaeae*, *C. fioriniae*, and *M. oryzae* were cultured under a 16 h black light/8 h dark cycle.

**Plant lines and growth**. *Arabidopsis thaliana* seeds were sown on rockwool (Grodan, Roermond, The Netherlands) and grown at 22 °C with 16 h of illumination per day in nutrient medium. *A. thaliana* accession Col-0 was used as the wild-type. *pen2-1*[3], *pen2-1 + PEN2::PEN2-GFP*[3], *pen3-1*[8], *edr1*[10], *edr1 pen2-1*[11], *gsh1-1 + GSH1::GSH1-GFP*[46], *pen2-1 gsh1-1*[46], *bak1-5*[75], *bik1-1 pbl1-1* (SALK_005291, SAIL_1236_D07)[29], *pepr1-1 pepr2-1* (SALK_059281, SALK_036564)[35] and *sid2-2*[76] mutants have been described previously. *gsh1-1* (CS3804), *chup1* (SALK_129128), *jac1* (SAIL_574_B09), *phot1* (SALK_088841), *phot2* (SALK_142275), *eds5-1* (CS3735) and *cas* (SALK_070416) mutants, and pt-ck (CS16265) and ER-ck (CS16250) organelle marker lines were obtained from Arabidopsis Biological Resource Center (Columbus, OH, USA).

**Plasmid construction**. *Escherichia coli* strain DH5α was used as the host for DNA manipulation. All primers used for plasmid construction are listed in Supplementary Table 1. All plasmids used for generating *A. thaliana* transgenic lines were derived from pRI101-AN, a vector for protein expression under the control of the cauliflower mosaic virus (CaMV) *35S* promoter (3262, TaKaRa Bio, Japan). The CaMV *35S* promoter sequence was removed from pRI-AtCHUP1p-AtCHUP1 to express *CHUP1* under its own promoter. For overexpression of CHUP1 protein, a fragment of *CHUP1* amplified from the *A. thaliana* genome was digested with NdeI and SmaI, and introduced into the corresponding site of pRI101-AN, resulting in pRI101-AtCHUP1. To generate CHUP1 with point mutations, the fragment of *CHUP1-R4A&S12A&R20A* was amplified and digested with NdeI and SmaI, and introduced into the corresponding site of pRI101-AN, resulting in pRI101-AtCHUP1-R4A&S12A&R20A. For complementation of the *chup1* mutant, the fragment containing the *CHUP1* gene with its upstream region, including the promoter sequence, was amplified from the *A. thaliana* genome, digested with PstI

and SmaI, and replaced with the corresponding region including the CaMV *35S* promoter in pRI101-AN, resulting in pRI-AtCHUP1p-AtCHUP1. For JAC1 protein overexpression, a fragment of *HA-JAC1* was amplified from the *A. thaliana* genome, digested with NdeI and EcoRI, and introduced into the corresponding site of pRI101-AN, resulting in pRI101-HA-AtJAC1. To generate C-terminally sfGFP-fused EDS5 and CAS, a fragment of *sfGFP* was amplified, digested with BamHI and EcoRI, and introduced into the corresponding site of pRI101-AN, resulting in pRI101-sfGFP-C. Next, SmaI-KpnI and NdeI-SmaI fragments of *EDS5* and *CAS*, respectively, were amplified from *A. thaliana* cDNA, digested with the indicated enzymes, and introduced into the corresponding sites of pRI101-sfGFP-C, resulting in pRI101-AtEDS5-sfGFP and pRI101-AtCAS-sfGFP, respectively.

For *CorbPLS1* gene disruption, an in vitro transposon tagging procedure was performed using pGPS-HYG-CAM[33]. A cosmid clone containing *PLS1* was used as the target. The gene disruption vector pKOPLS1 was constructed by mobilizing the modified Tn7 transposable element containing the hygromycin resistance gene *HPH* and chloramphenicol resistance gene cassette (GPS-HYG-CAM transposon) into the *PLS1* cosmid clone in vitro (Supplementary Fig. 4). The sequences of the transposon-inserted regions were determined using the primers DCPLS1s and DCPLS1as. In pKOPLS1, the transposon was inserted in the position between 23–24 bp (amino acid residue 8) of *PLS1* (Supplementary Fig. 4).

**Fungal transformation and gene disruption.** Protoplast preparation and transformation with transposon tagging were performed as described previously[33]. In brief, to obtain protoplasts, a cultured mycelium was harvested and treated with an enzyme solution containing 10 mg/ml of Driselase (D8037, Sigma-Aldrich) and 10 mg/ml of lysing enzyme from *Trichoderma harzianum* (L1412, Sigma-Aldrich) in 1.2 M $MgSO_4$ and 10 mM $Na_2HPO_4$ for 1.5 h. For transformation, the mixture of protoplasts and plasmid DNA was incubated on ice, and then gradually treated with 40% polyethylene glycol (PEG) solution at room temperature. After removal of PEG solution, protoplasts were poured onto a selection plate. Targeted gene disruption of *PLS1* was performed in *C. orbiculare* using the *CorbPLS1* gene disruption vector pKOPLS1 (Supplementary Fig. 4). Hygromycin-resistant transformants were selected on plates containing 100 μg/mL of hygromycin B (Wako Pure Chemicals, Japan).

**Generation of *A. thaliana* transgenic lines.** *Arabidopsis* transgenic plants were generated by *Agrobacterium*-mediated floral dipping transformation. The plasmids (pRI101-AN derivatives) were introduced into the *Agrobacterium tumefaciens* strain GV3101::pMP90 by electroporation. The cells of the bacterial transformants were cultured and harvested by centrifugation and suspended in a dipping solution (5% sucrose and 0.05% Silwet L-77). *A. thaliana* plants (5–6 weeks old) were used for the transformation. Transgenic plants were selected using kanamycin. Protein extracts were analyzed by immunoblotting using anti-CHUP1-H1 (αH)[42] and anti-HA (3F10, Roche) antibodies. HRP-linked anti-rabbit and anti-Rat IgG (NA934 and NA935, GE Healthcare) were used as secondary antibodies.

**Genotyping of *Arabidopsis* mutants.** The primers used for genotyping are listed in Supplementary Table 2. The *pen2-1*, *edr1-1*, *gsh1-1*, and *eds5-1* mutations were checked with the corresponding specific primers for the derived cleaved amplified polymorphic sequence (dCAPS) markers using dCAPS Finder 2.0 (http://helix.wustl.edu/dcaps/dcaps.html), and the PCR products (wild-type or mutant) types were cleaved with their corresponding restriction enzymes. Transfer DNA insertions into *CHUP1*, *JAC1*, *PHOT1*, *PHOT2*, and *CAS* genes were checked using specific primers. The fast-neutron-generated *sid2-2* mutation was also checked using the corresponding specific primers. The following plants with multiple mutations were generated via genetic crosses of each mutant: *edr1-1 pen2-1 gsh1-1*; *pen2-1 CHUP1*ox; *edr1-1 pen2-1 CHUP1*ox; *edr1-1 pen2-1 gsh1-1 CHUP1*ox; *pen2-1 chup1*; *edr1-1 pen2-1 chup1*; *edr1-1 pen2-1 gsh1-1 chup1*; *phot1 phot2*; *pen2-1 eds5-1*; *edr1-1 pen2-1 eds5-1*; *pen2-1 gsh1-1 eds5-1*; *edr1-1 pen2-1 gsh1-1 eds5-1*; *pen2-1 cas*; *edr1-1 pen2-1 cas*; *pen2-1 gsh1-1 cas*; *edr1-1 pen2-1 gsh1-1 cas*; *edr1-1 pen2-1 eds5-1 cas*; *pen2-1 gsh1-1 eds5-1 cas*; *edr1-1 pen2-1 gsh1-1 eds5-1 cas*; *edr1-1 pen2-1 gsh1-1 cas chup1*; *edr1-1 pen2-1 gsh1-1 eds5-1 cas chup1*; and *pen2-1 sid2-2*. The following plants expressing fluorescently-labeled immune proteins were generated via genetic crosses of each mutant: *gsh1 CHUP1*ox expressing *GSH1::GSH1-GFP*; *gsh1 chup1* expressing *GSH1::GSH1-GFP*; *CHUP1*ox expressing *CaMV 35S::EDS5-sfGFP*; *chup1* expressing *CaMV 35S::EDS5-sfGFP*; *CHUP1*ox expressing *CaMV 35S::CAS-sfGFP*; and *chup1* expressing *CaMV 35S::CAS-sfGFP*.

**Fungal inoculation.** For the pathogenicity assay of *Colletotrichum* fungi on *Arabidopsis*, a conidial suspension of each *Colletotrichum* fungus ($5 \times 10^5$ conidia/mL) was inoculated onto leaves of *Arabidopsis* seedlings and incubated for 7 days. For more fungal inoculum on the *chup1* mutant, a conidial suspension of *C. fioriniae* ($1.5 \times 10^6$ conidia/mL) was prepared. To measure the rates of fungal entry, ECR, surface localization of fluorescently labeled immune components, and surface localization of nuclei, the pathogen was inoculated on cotyledons. The inoculated cotyledons were mounted in water under a coverslip, with the inoculated surface facing the objective lens. The fungal entry rate (%) was calculated using the following numerical formula: (the number of melanized appressoria with formation

of invasive hypha)/(the number of melanized appressoria). The ECR rate (%) was calculated using the following numerical formula: (the number of epidermal cells with surface chloroplasts)/(the number of epidermal cells that contact the melanized appressorium). The ratio (%) of GSH1-GFP, EDS5-sfGFP, and CAS-sfGFP at the epidermal surface was calculated using the following numerical formula: (the number of epidermal cells with surface fluorescence of each immune component)/(the number of epidermal cells that contact the melanized appressorium). The ratio (%) of nuclei at the epidermal surface was calculated using the following numerical formula: (the number of epidermal cells with surface nuclei)/(the number of epidermal cells that contact the melanized appressorium).

**Bacterial inoculation.** For ECR analysis, a cell suspension of *Pst* DC3000 in distilled water (DW) ($OD_{600} = 1.0$) was drop-inoculated onto cotyledons of Col-0 plants and incubated. The ECR rate was calculated at 1, 2, and 3 days post inoculation (dpi). For the infection assay, a cell suspension of *Pst* DC3000 in 10 mM $MgCl_2$ ($OD_{600} = 1.0$) was drop-inoculated onto cotyledons of Col-0 plants and incubated for 4 days. The rate of bacterial growth inside the cotyledons was measured by enumeration of colonies on agar plates.

**RT-qPCR.** RT-qPCR was performed using the primers listed in Supplementary Table 3. The 11-day-old plants were used for fungal inoculation onto cotyledons. The inoculated plants were collected from at least five different plants of each genotype at 24 and 72 hpi. Total RNA was isolated from these plants using an RNeasy Plant Mini Kit (Qiagen) and treated with DNase (Qiagen) to remove any DNA contamination. A TaKaRa PrimeScript RT reagent Kit (Perfect Real Time) was used to obtain cDNA. The *Actin2* gene (At3g18780) was used as a control to normalize the cDNA levels. TaKaRa TB Green Premix Ex Taq II (Tli RNaseH Plus) was used for all RT-qPCR. Quantitative analysis of each mRNA was performed using a Thermal Cycler Dice Real Time System III TP970 (TaKaRa).

**Chemical treatments.** The chemical treatments used in this study were as follows. To stain callose deposition, each sample was visualized using 0.01% aniline blue fluorochrome solution (100-1, Biosupplies, Australia) with 0.07 M $K_2HPO_4$ after incubation for 2 h. For nuclear staining, each sample was soaked in 100 μg/mL DAPI solution (340-07971, Dojindo Laboratories, Japan) for 2 h. After incubation, the stainined samples were washed three times with DW. The inoculated cotyledons were observed using confocal laser-scanning microscopy. For trypan blue (TB) staining, the inoculated plants were soaked and incubated in lactophenol with TB (10 mL lactic acid, 10 mL glycerol, 10 mL phenol, 10 mL DW, and 10 mg TB) for 3 min at 95 °C. The plants were then transferred to a chloral hydrate solution for destaining. To detect the cell death of both epidermal and mesophyll cells, stained plants were incubated in chloral hydrate solution for 45 min. The inoculated cotyledons were mounted on a slide in 50% glycerol and observed under a light microscope (ECLIPSE E100, Nikon).

**Confocal laser-scanning microscopy.** To assess the fluorescent signals in *A. thaliana* and *Colletotrichum* fungi, the inoculated cotyledons on a glass slide were observed by confocal laser-scanning microscopy. The fluorescence was detected using an IX81 confocal microscope (Olympus) equipped with a diode laser (405/473/635 nm), LDD559 laser (559 nm), and a 60x UPlanSApo (1.35 numerical aperture) oil-immersion objective. Images were acquired and processed using FLUOVIEW FV1000-D (Olympus), Photoshop (Adobe), and ImageJ software (rsb. info.nih.gov/ij/). To detect DAPI, CFP, GFP/sfGFP, mCherry, and chlorophyll signals, fluorescence filters for DAPI, ECFP, EGFP, RFP, and Cy5 were used, respectively. The dichroic mirror, beam splitter, and emission filter sets were DM405/473/559/635, SDM560, and BA425-475 for DAPI, BA460-500 for CFP, BA485-585 for GFP, BA570-670 for mCherry, and BA650-750 for Chlorophyll. Analyses of papillary callose deposition and time series of moving chloroplasts in epidermal cells were performed using an LSM710 confocal microscope (Zeiss) equipped with a diode laser (405 nm), HeNe laser (633 nm), and a 63x Plan Apochromat (1.4 numerical aperture) oil-immersion objective. Images were acquired and processed using ZEN (Zeiss), Photoshop (Adobe), and ImageJ software. Fluorescence filters for DAPI and chlorophyll A were used to detect the signals of papillary callose and chlorophyll, respectively. The excitation and emission bandwidths were 405 nm and 410–585 nm for callose deposition and 633 nm and 647–721 nm for chlorophyll, respectively.

**Statistical analyses.** All statistical analyses were performed using R version 3.5.2. Levene's test was applied to check for heteroscedasticity between the treatment groups. Data were arcsine-transformed where necessary. To examine the differences among experimental groups, data were analyzed with Tukey's honestly significant difference (HSD test) and Dunnett's test, as appropriate. Differences were considered significant at $P < 0.05$.

## Data availability
The data supporting the findings of this work are included in the paper and its Supplementary Information files or are available from the authors upon reasonable request. Source data are provided with this paper.

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

## Acknowledgements

We thank Paul Schulze-Lefert for *pen2-1* and *pen2-1* with PEN2-GFP, Roger W. Innes for *edr1-1*, Yasuhiro Kadota for *bik1 pbl1*, Yube Yamaguchi for *pepr1-1 pepr2-1*, Volker Lipka for *sid2-2*, and Masamitsu Wada and Sam-Geun Kong for the anti-CHUP1 antibody. We thank Daisuke Shiomi for his technical assistance and helpful discussions. We thank Kei Hiruma, Asuka Hagiwara, Kazuho Takesue, Kosei Shiratori, and Hideki Baba for their technical assistance. We thank Katsuharu Saito for his help with statistical analyses. We also thank the Research Center for Supports to Advanced Science, Shinshu University for use of facilities. This work was supported by JSPS KAKENHI Grant Numbers 15K18648 (to H.I.), 18K05643 (to H.I.), and 18H02204 (to Y.T.), by the Asahi Glass Foundation (to Y.T.), and by the Leading Initiative for Excellent Young Researchers (LEADER) program of MEXT (to H.I.).

## Author contributions

H.I. designed and performed the research, analyzed the data, and wrote the article. Y.T. analyzed the data and wrote the article. H.I. supervised the project.

## Competing interests

The authors declare no competing interests.
