## [Peer Review File · Nature Communications]

REVIEWER COMMENTS

Reviewer #1 (Remarks to the Author):

This is an interesting manuscript and describes the role of epidermal chloroplasts in NHR defense against fungal pathogens. Although many of the findings agree with the conclusions, *chup1* and CHUP1ox data on ECR and infection does not correlate. Since *gsh1*, *eds5*, and *cas* mutants exhibited normal ECR, it is unclear on how these contribute to ECR and NHR. All bar graphs shown in all figures – need statistical significance values.

Data shown in Figure 1B and 1C clearly demonstrate the ECR against nonadapted fungi. Data using *pen2* mutant clearly shows that PEN2-mediated immunity is dominant over ECR and absence of *pen2* induces ECR.

In Figure 1D; authors should show Col-0 alone images to show the difference between wildtype and *pen2* in terms of fungal infection. In Figure 1C and 1E – there are no statistical significance values regarding the data?

Authors data on several mutants compromised for PAMP and DAMP perception or signaling, clearly indicate that ECR is not dependent on BAK1, BIK1 or PEPRs. Regarding the effector signal shown in Figure 2B images; authors should show control Col-0 to show that the signal is enhanced in *pen2* mutants. It is also unclear what we are looking at in Figure 2B microscope images. It will be good to show a few more fungal cells to show that it is not a single fungal cell in the cell periphery. These results are confusing because *sec22* deletion is supposed to decrease effector secretion; but the signal in the *sec22* deletion panel seems to be higher than Corb WT in the microscope image? In Figure 2 also, there are no statistical significance values shown in all bar graphs.

Figure 3 data: Authors state that ECR did not occur in *chup1* plants. This could be due to constitutive ECR observed in these plants; did authors use more fungal inoculum to see if the level of ECR could be enhanced in this background? Authors state that *jac1* mutant showed normal ECR; but in Figure 3C, *Cfio* and to some extent *Csia* seems to still induce ECR? In CHUP1 (R4A&S12A&R20A) lines, constitutive ECR seem to be lower than *chup1* mutant (based on the number of small chloroplasts near the membrane) – did author test in this background ECR could be induced upon infection with fungal pathogens? All graphs in Figure 3 need statistical significance values.

ECR role in NHR against fungal infection: Authors state that “the CHUP1ox and *chup1* single mutants retained normal resistance (Fig. 4A)”. This is somewhat surprising given *chup1* plants show constitutive ECR and CHUP1ox plants show loss of ECR upon fungal infection (Fig. 3). Based on these results, the expectation is that *chup1* plants should show enhanced resistance and CHUP1ox should show loss of resistance against fungal infection. Authors need to clarify these contradictory findings.

Figure 5 data: In addition to the bottom images shown in Figure 5A, the authors should show surface images for the control. Similarly, authors should show bottom images for comparison with surface images shown in Figure 5A upon infection with *Cfio*; this is important for the readers to see the differences. From the images it seems very few small chloroplasts are observed on the surface upon infection? I would have expected more chloroplasts based on the data shown in Figure 1.

Authors state regarding Figure 5B data: “The *eds5* and *cas* mutants retained normal resistance against these *Colletotrichum* fungi (Fig. 5B). Furthermore, *gsh1*, *eds5*, and *cas* mutants exhibited normal ECR (Supplementary Fig. 9). Remarkably, these three mutations all significantly reduced preinvasive NHR against *Cfio* and *Csia* in the *pen2* background (Fig. 5B)”.

These results are confusing; if *gsh1*, *eds5*, and *cas* exhibit normal ECR; then they should not have any role in fungal resistance as observed by the authors in Fig. 5B. Then how GSH1, EDS5 and CAS contribute to ECR and NHR?

In Figure 7A – Col-0 control; where is the nucleus?

Reviewer #2 (Remarks to the Author):

This manuscript by Irieda and Takano presents the interesting observation that epidermal chloroplasts move to the surface of the cell in contact with the environment (a phenomenon for which the authors coin the term “epidermal chloroplast response”, or ECR) upon attempted penetration by non-adapted fungi. The authors then demonstrate that this response depends on fungal secretion, that it requires known regulators of chloroplast movement in response to light cues, and, by generating multiple mutant combinations, that it may contribute to non-host resistance. This work also identifies three known immune regulators localized in chloroplasts, GSH1, EDS5, and CAS, as contributors to non-host resistance. Finally, the authors show that the nucleus also moves to the epidermal surface upon fungal attack, and that this re-localization requires wild-type chloroplast movement.

Overall, the interesting and intriguing results presented here support a particular role of epidermal chloroplasts in plant defence. Nevertheless, I have the feeling that some of the main conclusions of the paper are based on assumptions, and presented as over-statements.

One important question is whether the effect of loss-of-function/overexpression of CHUP/JAC1 on non-host resistance is direct or indirect. Does altered function of these regulators affect susceptibility to apoplastic pathogens (e.g. *Pseudomonas syringae* strains inoculated into the leaf)? Does altered photorelocation affect fungal penetration (both in the WT or the *pen2* backgrounds)? Another aspect that needs to be clarified is whether the apparent increased accumulation of the analyzed defence regulators (GSH1, EDS5, CAS) in epidermal chloroplasts is due to the intrinsic differences between epidermal and mesophyll chloroplasts (in number and size). Are similar differences in accumulation observed for other chloroplast-localized proteins not involved in defence? Is the overall accumulation of proteins expressed from a 35S promoter different between epidermal and mesophyll cells?

Additional comments:

- Lines 145-146: The authors only test the involvement of PEPR1/2; therefore, the conclusion should be more specific/toned down.
- Lines 276-277: The authors do not have enough experimental evidence to reach this conclusion – stromal proteins localize in stromules, but that does not imply they are transported. This applies again to line 458.
- Lines 470-471: Transport of small signalling molecules and immune-related protein where?
- Figure 5A: I would suggest that the authors show the GFP channel only as well.
- Figure 6C: Why did the authors not test the contribution of CHUP1 here?

Minor comments:

- Line 61: This would be “proposed” rather than “revealed”.
- There are a number of recent references that the authors may consider including in the manuscript, particularly Toufexi et al., 2019, bioRxiv, and Ding et al., 2019, MPP.
- The authors should explain in the text what *cst1*, *pls1*, *atg26*, and *icl1* are.
- Line 136: PAMPs (pl).
- Line 254: Are the GSH1-GFP-expressing plants in the *gsh1* background? This is not indicated in the figure.
- Line 271: “not” is missing?
- Line 305: Does Ref. 53 show this?
- Line 316: “constitute the immune system” sounds way too strong – please tone down.
- English language needs editing throughout the text.
- In my opinion, title and abstract do not clearly convey the conclusions of the work.

Reviewer #3 (Remarks to the Author):

The manuscript entitled 'Epidermal chloroplast is a motile guardian equipped with plant immune components' is interesting in that the authors found the requirement of epidermal chloroplasts for plant nonhost resistance to nonadapted *Colletotrichum* fungi. By genetic and cell biological

approaches, the authors reached the conclusion that emerged epidermal chloroplasts in response to fungal attempts comprise an additional, likely lower-level, immune system to the PEN2/PEN3 pathway. The previously unrevealed importance of epidermal chloroplasts for plant immunity can clearly attract an interest from many plant scientists. However, the authors should address the below issues to support their claim.

1. The authors tested bak1, bik1 and pepr1 pepr2 mutants for finding an ECR cue, and failed. But, based on the failure to induce ECR by mutant fungi not to form pegs, it is suggested that a mechanical stimulus is likely to induce ECR. Indeed, it was reported that application of a mechanical stimulus induces a subset of plant immune responses (PNAS 95: 8398). Therefore, the authors should test whether mechanical wounding itself can induce ECR.
2. The authors found that Cfio induces some defense-related genes in pen2 mutant but not in WT. They explained this with a threshold concept that ECR level is low in WT to induce those genes. However, in Fig. 1C, ECR level in WT at 3 dpi is comparable to that in pen2 mutant at 1 and 2 dpi. Then, those genes are induced at 3 dpi in WT?
3. Since there are many multiple mutants, it is understandable for the authors to omit some controls (WT and respective single mutants) in most figures showing fungal entry rates and gene expression. However, for readability, the authors should show all these controls.
4. In some mutants shown in Supp Fig 11, fungal entry rates of Cfio and Csia reached almost 90%. These mutants can also allow more penetration of an adapted fungus. How is the susceptibility of Chig in those mutants? Indeed, ECR to Chig is significantly elevated in pen2 mutant in Fig. 1C.
5. The authors argued that the ECR system might work as a lower-level mechanism to the PEN2/PEN3 system. Then, the authors should consider why plants have this additional immune system and when the PEN2/PEN3 pathway can be disrupted during plant immune responses.
6. In lines 434-436 on page 20, maybe 'increased' instead of 'decreased'?

Reviewer #4 (Remarks to the Author):

The manuscript „Epidermal chloroplast is a motile guardian equipped with plant immune components“ reports on novel discovery that chloroplast movement contributes to leaf epidermal immunity to fungal invasion. This observation is particularly relevant for pen2-immuno-compromised Arabidopsis genotypes in interaction with directly cell wall penetrating fungi. Cell biological studies are supported by genetic evidence that chloroplast movement/recruitment to the cell surface actively contributes to pre-invasive immunity. Little is known about the contribution of chloroplasts in general to plant immunity. This is even more true for small epidermal chloroplasts, which are generally under-explored in epidermal plant cell biology. I therefore think the contribution gives interesting and perhaps surprising insight into a new function of chloroplasts in plant immunity. I enjoyed reading and found data largely convincing. However, I have several points to mention, which, I feel, would contribute to further improve the study/manuscript and make conclusions more precise.

General comments:

I wonder whether the authors have some evidence, that ECR would be also of importance in a wilt type background because basically all results are in the pen2 background. I see that this is perhaps necessary to have an immune-compromised background to see the patho-phenotypes, but there is a certain concern, that we are looking a pen2 pleiotropic effects, which must not come to action in a more natural scenario. Another way to show this would be testing other pen mutants.

The CHUP mutants affect cell entry of parasitic fungi but hardly affect expression of defense genes. It therefore remains unclear how ECR could contribute to pre-invasive immunity. This could be explained by diverse mechanisms, which would be more or less specific for the actual host immune response. The conclusions that epidermal chloroplasts act in immunity by positioning immune (from the abstract “motile guardians specifically accommodating immune regulators in plant epidermis”) components during the host response is therefore tempting. However, were is actually the direct evidence for this? I believe, to prove this, you would need genetic evidence, that CAS,

GSH1 and EDS5 act genetically redundantly with CHUP. This, however, appears not to be the case when looking at Figure 5E. Therefore I am not fully convinced that ECR is functionally linked to or acts "cooperatively" with the other components introduced here (GSH; EDS5, EDR1, CAS).

I think the use of statistical testing is a bit selective over the entire data set. I guess the manuscript would profit from general ANOVA and post hoc testing in all data panels or from other statistical methods with correction for multiple testings.

I wonder whether authors demonstrated ECR by any other means than chloroplast autofluorescence, which, in the way it was recorded, could be also autofluorescence of substances other than chlorophyll.

Other comments:

Line 23: ECR-fixed: I wonder whether this is a good term? Perhaps consider constitutively activated ECR

Line 27: what do you mean with accommodating? Please be more precise here!

Line 48: perhaps consider mentioning what type of transporter PEN3 is.

Line 59: chloroplasts do not produce calcium: Do you mean release?

Line 92: ECR does not enhance nonhost resistance but is required for, or involved in, or functions in.

Line 163: I am actually not sure whether you can conclude this based on a correlation. The delta_sec22 mutant certainly has also a virulence defect and therefore less fungal progression may simply cause less ECR response.

Figure 5a:

The localization of GSH1, EDS5 etc. on small epidermal chloroplasts and at stomules looks specific to me. However, I wonder whether we are looking at z-stacks here. If not, please add the entire z-stack from the same cell so that the reader can distinguish specific localization from cytoplasmic background (CAM35S-over-expressed?). Also mention in the figure legend, which promoter was used for expression.

Line 316: To me, data point to additive rather than cooperative functions.

Line 340: What do you mean by guide? The wording appears a bit inappropriate to me.

Line 361: Nuclear movement is CHUP-dependent not ECR dependent: Please be more precise, at least in the results part.

Line 388: "penetration-specific". This could also be mechanical triggers, right?

Line 390-91: I do not agree. I rather see an indirect effect of the sec22 mutation. Also, if an effector would be recognized we would expect it to trigger a hypersensitive reaction, which is a hallmark of plant effector-triggered immunity.

Line 403-405: To my understanding, gene expression and ECR were recorded at different points in time and should therefore not compared to each other.

Line 445: I think there is a lot of literature that shows nucleus translocation to sites of fungal attack. Please consider discussing this.

Line 457-458: Your data do not show transport by but localization at stromules.

Line 468: There is only one immune system, to which ECR might contribute.

Line 474: What do you mean with guardians. Actually, you do not show intrinsic function of chloroplasts themselves but rather of other components that hitchhike on them. I personally would appreciate a clearer and less projecting wording of your findings.

Responses to reviewer 1:

//Reviewer 1 comment:

Reviewer #1 (Remarks to the Author):

*This is an interesting manuscript and describes the role of epidermal chloroplasts in NHR defense against fungal pathogens. Although many of the findings agree with the conclusions, *chup1* and *CHUP1ox* data on ECR and infection does not correlate. Since *gsh1*, *eds5*, and *cas* mutants exhibited normal ECR, it is unclear on how these contribute to ECR and NHR. All bar graphs shown in all figures – need statistical significance values.*

//Author reply:

Thank you for your critical reading of our manuscript and insightful comments, which have helped us to improve our manuscript considerably. Based on the comments provided by the reviewers, we revised our manuscript after performing new experiments. We have addressed all the reviewers' concerns. In particular, we provided substantial responses to the following points, in addition to all other comments: i) the relationship between overexpression/mutation of CHUP1 and antifungal preinvasive nonhost resistance (NHR) in *Arabidopsis*; and ii) the link between immune components (GSH1, EDS5, and CAS) and the ECR in *Arabidopsis* NHR. Please see our point-by-point responses to each comment. We also performed statistical tests in all bar graphs shown in all figures and revised the Methods section accordingly (lines 783–788).

//Reviewer 1 comment:

*Data shown in Figure 1B and 1C clearly demonstrate the ECR against nonadapted fungi. Data using *pen2* mutant clearly shows that PEN2-mediated immunity is dominant over ECR and absence of *pen2* induces ECR.*

*In Figure 1D; authors should show Col-0 alone images to show the difference between wildtype and *pen2* in terms of fungal infection. In Figure 1C and 1E – there are no statistical significance values regarding the data?*

//Author reply:

As the reviewer suggested, we added images of Col-0 alone for comparison with the *pen2* images in Fig. 1D (New Fig. 1D).

Multiple tests were performed, as shown in Fig. 1C and 1E (New Fig. 1C, 1E).

//Reviewer 1 comment:

Authors data on several mutants compromised for PAMP and DAMP perception or signaling, clearly indicate that ECR is not dependent on BAK1, BIK1 or PEPRs. Regarding the effector signal shown in Figure 2B images; authors should show control Col-0 to show that the signal is enhanced in *pen2* mutants. It is also unclear what we are looking at in Figure 2B microscope images. It will be good to show a few more fungal cells to show that it is not a single fungal cell in the cell periphery. These results are confusing because *sec22* deletion is supposed to decrease effector secretion; but the signal in the *sec22* deletion panel seems to be higher than *Corb* WT in the microscope image? In Figure 2 also, there are no statistical significance values shown in all bar graphs.

//Author reply:

As the reviewer suggested, we added images of Col-0 for comparison with the *pen2* images (New Fig. 2B).

Corb sec22 deletion mutants show intracellular retention of effector signals due to membrane traffic defects (Irieda et al., 2014), resulting in a higher signal than *Corb* WT. Thus, we displayed a few more panels of the *sec22* mutant and added an additional explanation in the revised manuscript (lines 169–170), as follows: “and the effector signals were abnormally retained and dispersed inside fungal cells when $\Delta sec22$ was inoculated on host cucumber³⁸.”

Reference:

38. Irieda, H. et al. *Colletotrichum orbiculare* secretes virulence effectors to a biotrophic interface at the primary hyphal neck via exocytosis coupled with SEC22-mediated traffic. *Plant Cell* 26, 2265–2281 (2014).

We performed multiple tests in all figures in New Fig. 2.

//Reviewer 1 comment:

Figure 3 data: Authors state that ECR did not occur in *chup1* plants. This could be due to constitutive ECR observed in these plants; did authors use more fungal inoculum to see if the level of ECR could be enhanced in this background? Authors state that *jac1* mutant showed normal ECR; but in Figure 3C, *Cfio* and to some extent *Csia* seems to still induce ECR? In *CHUP1* (R4A&S12A&R20A) lines, constitute ECR seem to be lower than *chup1* mutant (based on the number of small chloroplasts near the membrane) – did author test in this background ECR could be induced upon infection with fungal pathogens? All graphs in Figure 3 need statistical significance values.

//Author reply:

As the reviewer suggested, we performed a new experiment on the ECR of the *chup1* mutant in

response to *Cfio* inoculation at a high concentration (three times) (New Fig. 3D, left). Statistical analyses showed no difference in the percentage of epidermal cells with surface chloroplasts with or without fungal inoculation, but additional quantification revealed that the number of surface chloroplasts in the epidermis increased to some extent (New Fig. 3D, right). Thus, we conclude that the *chup1* mutant still has the ability to slightly move epidermal chloroplasts. We added new sentences and revised the manuscript as follows (lines 202-209).

“A more fungal inoculum on the *chup1* mutant did not significantly increase the population of ECR-activated cells, but the number of surface chloroplasts was slightly increased (Fig. 3D). Therefore, the *chup1* mutant showed constitutive positioning of the epidermal chloroplasts at the surface, but the mutant is deficient in ECR that is newly triggered by the inoculation of nonadapted *Colletotrichum* fungi (Fig. 3B, 3C, 3D). Thus, both overexpression and mutation of CHUP1 commonly cause impairments in the intracellular movement of epidermal chloroplasts in the ECR.”

In accordance with reviewer’s comment on the ECR induction in the *jac1* mutant, we corrected the word “normal” to “near-normal” (lines 210, 520–521).

As the reviewer suggested, we performed a new quantification of the ECR in the *CHUP1-R4A&S12A&R20Aox* lines in response to *Cfio* (New Fig. 3E, lower graphs). As the reviewer suggested, the data showed that constitutive surface localization of epidermal chloroplasts was lower than that of the *chup1* mutant (New Fig. 3E), and this phenotype correlated with decreased levels of endogenous CHUP1 proteins (New Fig. 3A). Furthermore, the ECR was induced in these mutant lines (New Fig. 3E, lower graphs). Thus, we have added new sentences in the revised manuscript (lines 217–220).

We performed multiple tests in all figures in New Fig. 3.

//Reviewer 1 comment:

ECR role in NHR against fungal infection: Authors state that “the CHUP1ox and chup1 single mutants retained normal resistance (Fig. 4A)”. This is somewhat surprising given chup1 plants show constitutive ECR and CHUP1ox plants show loss of ECR upon fungal infection (Fig. 3). Based on these results, the expectation is that chup1 plants should show enhanced resistance and CHUP1ox should show loss of resistance against fungal infection. Authors need to clarify these contradictory findings.

//Author reply:

CHUP1ox plants had defective epidermal chloroplast movement (ECR) (Fig. 3B, New Fig. 3C). In contrast, the chup1 mutant constitutively fixes a specified population of epidermal chloroplasts at the

surface area and restricts their movement, thereby resulting in an ECR-deficient phenotype (Fig. 3B, New Fig. 3C). In terms of impairment of the CHUP1-associated moving ability of epidermal chloroplasts in the ECR, both mutants commonly exhibited an impaired ECR. To clarify these points, we added an additional explanation in the revised manuscript (Lines 207–209), as follows:

“Thus, both overexpression and mutation of CHUP1 commonly cause impairments in the intracellular movement of epidermal chloroplasts in the ECR.”

PEN2-mediated immunity is dominant over the ECR, and the absence of functional PEN2 induces a strong ECR in response to *Cfio*, *Csia*, and *Corb* (New Fig. 1C). Thus, the entry rate of these fungi into *CHUP1ox* and *chup1* plants, which normally retain the PEN2-related higher-layer preinvasive defense, is comparable to that of wild-type plants (New Fig. 4A). Consistently, in the *pen2* background, fungal entry rates of *Cfio* and *Csia* correlated with the impairments in the CHUP1-associated ECR (New Fig. 3C, New Fig. 4A, 4C, 4D). Furthermore, in the revised manuscript, we added a new nonadapted *Colletotrichum* fungus *Cnym*, which showed a somewhat higher entry rate into wild-type *Arabidopsis* than *Cfio* (New Fig. 8). Consistently, the *CHUP1ox* and *chup1* single mutants permitted higher amounts of *Cnym* entry compared to wild-type plants (New Fig. 8). Please see the new paragraph on the experiments using *Cnym* (lines 379–403). We have also revised the sentence in the relevant part of the Discussion section (lines 472–474).

//Reviewer 1 comment:

Figure 5 data: In addition to the bottom images shown in Figure 5A, the authors should show surface images for the control. Similarly, authors should show bottom images for comparison with surface images shown in Figure 5A upon infection with Cfio; this is important for the readers to see the differences. From the images it seems very few small chloroplasts are observed on the surface upon infection? I would have expected more chloroplasts based on the data shown in Figure 1.

//Author reply:

In accordance with the reviewer’s comment, we added surface images of each fluorescent marker plant line without fungal inoculation for the control (New Fig. 5, upper right). Similarly, we added bottom images with fungal inoculation (New Fig. 5, lower left). The images for Fig. 5 were taken at 2 dpi; thus, the number of chloroplasts at the surface is smaller compared to the 3 dpi image displayed in Fig. 1. Furthermore, as shown in Fig. 1, there are many chloroplasts clustered around the nucleus.

//Reviewer 1 comment:

Authors state regarding Figure 5B data: “The eds5 and cas mutants retained normal resistance against these Colletotrichum fungi (Fig. 5B). Furthermore, gsh1, eds5, and cas mutants exhibited

normal ECR (Supplementary Fig. 9). Remarkably, these three mutations all significantly reduced preinvasive NHR against *Cfio* and *Csia* in the *pen2* background (Fig. 5B)".

These results are confusing; if *gsh1*, *eds5*, and *cas* exhibit normal ECR; then they should not have any role in fungal resistance as observed by the authors in Fig. 5B. Then how *GSH1*, *EDS5* and *CAS* contribute to ECR and NHR?

//Author reply:

Our results show that *GSH1*, *EDS5*, and *CAS* do not participate in ECR regulation in terms of epidermal chloroplast movement. On the other hand, these three components were clearly involved in NHR in the *pen2* background (New Fig. 7A, 7D, New Supplementary Fig. 11). Importantly, we here revealed that these components preferentially localized to the epidermal chloroplasts (New Fig. 5, New Supplementary Fig. 8, New Supplementary Movie 2).

To analyze the relationship between the ECR and these components in more detail, we generated fluorescent marker plant lines of these immune components in the *CHUP1ox* and *chup1* backgrounds (New Fig. 6). We found that their localization patterns strongly correlated with epidermal chloroplasts and the ECR (New Fig. 5, New Fig. 6). Thus, the ECR clearly supports drastic changes in the subcellular locations of *GSH1*, *EDS5*, and *CAS* in response to fungal inoculation. Please see the new paragraph explaining the relationship between *GSH1*, *EDS5*, *CAS* locations, and the ECR (lines 299–304). We believe that the ECR plays supportive roles in the immune pathways of these components by moving chloroplasts in the epidermis.

This idea is further supported by our new experiments using the additionally identified fungus *Csia* (low-invasive strain), which showed lower invasion ability than *Csia* (New Fig. 7D) and low ECR induction in wild-type plants (New Supplementary Fig. 13). Importantly, *gsh1*, *eds5*, *cas* and ECR-deficient mutations showed redundant effects on NHR against *Csia* (low-invasive strain) (New Fig. 7D), strongly suggesting that the ECR acts in a genetically redundant manner with *GSH1*, *EDS5*, and *CAS* on the preinvasive NHR. Thus, the ECR is functionally linked to these immune components (*GSH1*, *EDS5*, and *CAS*). We also added a new paragraph on *Arabidopsis* NHR against *Csia* (low-invasive strain) (lines 356–377) and revised the sentence in the relevant part of the Discussion section (lines 543–546).

//Reviewer 1 comment:

In Figure 7A – *Col-0* control; where is the nucleus?

//Author reply:

Nuclei with clusters of epidermal chloroplasts cannot be detected at the surface in the steady state (without pathogen). Fig. 7A (New Fig. 10A) displays the surface images of epidermal pavement cells;

therefore, no nuclei were observed in the Col-0 control (+ DW). To avoid confusion, we added an additional explanation to the revised manuscript as follows (line 440). “whereas no nuclei were observed at the epidermal surface without fungal inoculation”

We have also added Supplementary Fig. 16 in the revised manuscript to visualize the summary of our manuscript.

Finally, we reiterate our thanks to Reviewer 1 for improving our manuscript.

Responses to reviewer 2

//Reviewer 2 comment:

Reviewer #2 (Remarks to the Author):

This manuscript by Irieda and Takano presents the interesting observation that epidermal chloroplasts move to the surface of the cell in contact with the environment (a phenomenon for which the authors coin the term “epidermal chloroplast response”, or ECR) upon attempted penetration by non-adapted fungi. The authors then demonstrate that this response depends on fungal secretion, that it requires known regulators of chloroplast movement in response to light cues, and, by generating multiple mutant combinations, that it may contribute to non-host resistance. This work also identifies three known immune regulators localized in chloroplasts, GSH1, EDS5, and CAS, as contributors to non-host resistance. Finally, the authors show that the nucleus also moves to the epidermal surface upon fungal attack, and that this re-localization requires wild-type chloroplast movement.

Overall, the interesting and intriguing results presented here support a particular role of epidermal chloroplasts in plant defence. Nevertheless, I have the feeling that some of the main conclusions of the paper are based on assumptions, and presented as over-statements.

//Author reply:

Thank you for your critical reading of our manuscript and your insightful suggestions and positive comments, which have helped us to improve our manuscript considerably. Based on the comments provided by the reviewers, we revised our manuscript after performing new experiments. We have addressed all of the reviewers' concerns. Please see our point-by-point responses.

//Reviewer 2 comment:

One important question is whether the effect of loss-of-function/overexpression of CHUP/JAC1 on non-host resistance is direct or indirect. Does altered function of these regulators affect susceptibility to apoplastic pathogens (e.g. Pseudomonas syringae strains inoculated into the leaf)?

//Author reply:

We agree with the reviewer; we also think it is important to clarify the direct link between the CHUP1/JAC1-regulated epidermal ECR and non-host resistance (NHR) against pathogenic fungi.

In the original manuscript, we showed that the immune components GSH1, EDS5, and CAS were preferentially localized to epidermal chloroplasts. We also showed that mutations in these immune components reduce NHR in the *pen2* background. Thus, we considered that the ECR regulates the location of these immune components, which contributes to NHR.

In the revised manuscript, to further address the link between the CHUP1-dependent ECR and these

immune components (GSH1, EDS5, and CAS) in the epidermis, we additionally generated fluorescent marker plant lines of these immune components in the *CHUP1ox* and *chup1* backgrounds (New Fig. 6). We found that their localization patterns strongly correlated with epidermal chloroplasts and the ECR (New Fig. 5, New Fig. 6). Thus, the CHUP1-dependent ECR clearly supports drastic changes in the subcellular locations of the immune components GSH1, EDS5, and CAS in response to fungal inoculation. Please see the new paragraph explaining the relationship between the locations of GSH1, EDS5, and CAS and the ECR (lines 299-304).

Furthermore, to address the functional link between the ECR and GSH1, EDS5, and CAS in epidermal NHR, we performed new experiments using the additionally identified fungus *Csia* (low-invasive strain), which showed lower invasion ability than *Csia* (New Fig. 7D) and low ECR induction in the wild-type plant (New Supplementary Fig. 13). Importantly, *gsh1*, *eds5*, *cas* and ECR-deficient mutations showed redundant effects on epidermal NHR against *Csia* (low-invasive strain) (New Fig. 7D), strongly suggesting that the ECR acts with GSH1, EDS5, and CAS on the preinvasive NHR in a genetically redundant manner. Thus, the ECR directly contributes to preinvasive NHR by controlling the location of immune components. We have added a new paragraph on epidermal NHR against *Csia* (low-invasive strain) (lines 356–377) and revised the sentences in the relevant part of the Discussion section (lines 543–546).

As the reviewer suggested, the susceptibility test using the bacterial pathogen *Pseudomonas syringae* is also suitable to further elucidate whether the loss-of-function/overexpression of CHUP1/JAC1 directly affects the ECR and epidermal NHR, because *P. syringae* cannot penetrate the epidermis but enters through natural openings (e.g., stomata) and multiplies in apoplastic space. Therefore, we performed additional experiments using *P. syringae* in the revised manuscript (New Fig. 9E, 9F, New Supplementary Fig. 15). We first checked whether the ECR was induced after inoculation with *P. syringae* (New Fig. 9E, New Supplementary Fig. 15). As expected, the ECR did not occur against *P. syringae*, further supporting the notion that ECR triggers are specific to the fungal penetration stage. Next, we assessed the susceptibility of *CHUP1ox*, *chup1*, *HA-JAC1ox*, and *jac1* plants to *P. syringae* (New Fig. 9F). Consistent with the lack of ECR induction, the susceptibility of ECR-impaired plants to *P. syringae* was comparable to that of the wild-type plants. Thus, we have added new sentences in the revised manuscript (lines 417–422).

//Reviewer 2 comment:

Does altered photorelocation affect fungal penetration (both in the WT or the pen2 backgrounds)?

//Author reply:

CHUP1 is a regulator engaged in both photorelocation of mesophyll chloroplasts (Oikawa et al., 2003, 2008) and the ECR of epidermal chloroplasts (New Fig. 3). However, the *chup1* single mutation did not affect the penetration of *Cfio*, *Csia*, and *Corb* (New Fig. 4A), because higher-layer PEN2-related NHR was still working sufficiently. Therefore, it is assumed that photorelocation-specific single mutations (e.g., *phot2*, Iwabuchi et al., 2010) also have no effect on the penetration of *Cfio*, *Csia*, and *Corb*. We did not have a *phot2* mutant in the *pen2* background, so we were not able to evaluate the effect of *phot2* mutation on fungal penetration of *Cfio*, *Csia*, and *Corb*. Alternatively, we have identified an additional nonadapted *Colletotrichum* strain *Cnym* with a somewhat higher entry rate into wild-type *Arabidopsis* than *Cfio* (New Fig. 8). The entry rate of *Cnym* into the photorelocation-specific *phot2* single mutant was comparable to that in wild-type plants; meanwhile, ECR-impaired mutants permitted increased entry of *Cnym* (New Fig. 8C, 8D). Thus, we believe that altered photorelocation does not affect fungal penetration. Please see the new paragraph on the experiments using *Cnym* (lines 379–403).

References:

40. Oikawa, K. et al. CHLOROPLAST UNUSUAL POSITIONING1 Is Essential for Proper Chloroplast Positioning. *Plant Cell* 15, 2805-2815 (2003).
42. Oikawa, K. et al. Chloroplast Outer Envelope Protein CHUP1 Is Essential for Chloroplast Anchorage to the Plasma Membrane and Chloroplast Movement. *Plant Physiol.* 148, 829-842 (2008).
59. Iwabuchi, K. et al. Actin Reorganization Underlies Phototropin-Dependent Positioning of Nuclei in *Arabidopsis* Leaf Cells. *Plant Physiol.* 152, 1309-1319 (2010).

//Reviewer 2 comment:

Another aspect that needs to be clarified is whether the apparent increased accumulation of the analyzed defence regulators (GSH1, EDS5, CAS) in epidermal chloroplasts is due to the intrinsic differences between epidermal and mesophyll chloroplasts (in number and size). Are similar differences in accumulation observed for other chloroplast-localized proteins not involved in defence? Is the overall accumulation of proteins expressed from a 35S promoter different between epidermal and mesophyll cells?

//Author reply:

It has already been reported that EDS5 is preferentially localized in epidermal chloroplasts rather than mesophyll chloroplasts when expressed under both its own promoter and 35S promoters (Yamasaki et al., 2013). Furthermore, the mesophyll photorelocation regulator CHUP1 is expressed under its own promoter, localized both in epidermal and mesophyll chloroplasts (Higa et al., 2014). This localization pattern of CHUP1 is clearly different from that of GSH1 expressed from its own promoter and that of

EDS5 (New Fig. 5, New Fig. 6, New Supplementary Fig. 8, Yamasaki et al., 2013). Therefore, we added an additional explanation as follows in the revised manuscript (lines 292–293), as follows: “These localization patterns are different from those of CHUP1, which localizes to both epidermal and mesophyll chloroplasts⁵⁴.”

GSH1 promoter-driven GSH1-GFP protein accumulated predominantly in epidermal chloroplasts compared to in mesophyll chloroplasts, suggesting preferential GSH1 expression in the epidermis (New Fig. 5, New Supplementary Fig. 8, New Supplementary Movie 2). We did not have a plant line expressing GSH1-GFP under the *35S* promoter; therefore, there is no information about its localization patterns in epidermal and mesophyll chloroplasts. On the other hand, EDS5-GFP expressed under the *35S* promoter showed a similar localization pattern to *EDS5* promoter-driven EDS5-GFP; it is specific to the chloroplast periphery both in epidermal and mesophyll cells, although the signal intensity was much higher in epidermal chloroplasts (Yamasaki et al., 2013). This chloroplast periphery-specific pattern was also observed in our transgenic plant line expressing EDS5-sfGFP expressed under the *35S* promoter (New Fig. 5, New Supplementary Fig. 8, New Supplementary Movie 2). In a previous report, CAS-GFP expressed under the *35S* promoter showed similar localization patterns in both epidermal and mesophyll chloroplasts (Nomura et al., 2008). In our experiment, the localization pattern of *35S* promoter-driven CAS-sfGFP in epidermal chloroplasts was also the same, although the signal intensity was much higher in epidermal chloroplasts in this study (New Fig. 5, New Supplementary Fig. 8, New Supplementary Movie 2). To explain more precisely, we added additional sentences explaining the same localization patterns of these components in the revised manuscript (lines 271–275):

“When only expressed under the CaMV *35S* promoter, mesophyll chloroplasts had weak EDS5-GFP signals, but the chloroplast periphery-specific pattern was maintained in two types of chloroplasts⁴⁹. CAS-GFP driven by the CaMV *35S* promoter was detected in both epidermal and mesophyll chloroplasts, with no difference in their localization patterns⁵⁰.”

Reference:

50. Yamasaki, K. et al. Chloroplast envelope localization of EDS5, an essential factor for salicylic acid biosynthesis in *Arabidopsis thaliana*. *Plant Signal. Behav.* **8**, e23603 (2013).
55. Higa, T. et al. Actin-dependent plastid movement is required for motive force generation in directional nuclear movement in plants. *Proc. Natl. Acad. Sci. USA* **111**, 4327-4331 (2014).
51. Nomura, H. et al. Evidence for chloroplast control of external Ca²⁺-induced cytosolic Ca²⁺ transients and stomatal closure. *Plant J.* **53**, 988-998 (2008).

//Reviewer 2 comment:

Additional comments:

- Lines 145-146: *The authors only test the involvement of PEPRI/2; therefore, the conclusion should be more specific/toned down.*

//Author reply:

As the reviewer suggested, we toned down the expression and revised the sentence in the revised manuscript as follows: “Combined with the fact that BAK1 and BIK1 also function in Pep1 DAMP signaling^{36,37}, Pep1 and its paralogs might be not a cue of ECR.” (lines 160–161).

//Reviewer 2 comment:

- Lines 276-277: *The authors do not have enough experimental evidence to reach this conclusion – stromal proteins localize in stromules, but that does not imply they are transported. This applies again to line 458.*

//Author reply:

In accordance with the reviewer’s comment, we have toned down the expression and revised the sentences as follows:

“Therefore, our findings suggest that the ECR may at least partially regulate the localization of *Arabidopsis* immune-related proteins via stromules.” (lines 312–314)

“Therefore, our data indicate that stromules may potentially transport immune-related *Arabidopsis* proteins.” (lines 579–580)

//Reviewer 2 comment:

- Lines 470-471: *Transport of small signalling molecules and immune-relate protein where?*

//Author reply:

In accordance with the reviewer’s comment, we have changed the words “transport” to “intracellular traffic” in the revised manuscript (line 592).

//Reviewer 2 comment:

- *Figure 5A: I would suggest that the authors show the GFP channel only as well.*

//Author reply:

As the reviewer suggested, we have added the GFP channel images (New Fig. 5).

//Reviewer 2 comment:

- Figure 6C: Why did the authors not test the contribution of CHUP1 here?

//Author reply:

As the reviewer pointed out, we added the fungal entry data on the *chup1* and *pen2 chup1* mutants (New Fig. 9C).

//Reviewer 2 comment:

Minor comments:

- Line 61: This would be “proposed” rather than “revealed”.

//Author reply:

In accordance with the reviewer’s comment, we have replaced the word “revealed” to “proposed” in the revised manuscript (line 63).

//Reviewer 2 comment:

- There are a number of recent references that the authors may consider including in the manuscript, particularly Toufexi et al., 2019, bioRxiv, and Ding et al., 2019, MPP.

//Author reply:

We thank the reviewer for providing valuable information. As the reviewer suggested, we added new sentences including relevant information from these references in the Discussion section (lines 553–562).

References:

67. Toufexi, A. et al. Chloroplasts navigate towards the pathogen interface to counteract infection by the Irish potato famine pathogen. bioRxiv 516443, Preprint at <https://doi.org/10.1101/516443> (2019).

As the reviewer suggested, we added the following reference in the revised manuscript (line 436).

References:

60. Ding et al. Chloroplast clustering around the nucleus is a general response to pathogen perception in *Nicotiana benthamiana*. Mol. Plant Pathol. **20**, 1298-1306 (2019).

//Reviewer 2 comment:

- The authors should explain in the text what *cst1*, *pls1*, *atg26*, and *icl1* are.

//Author reply:

In accordance with the reviewer's comment, we have added explanations about these *Corb* mutants and revised the relevant sentences (lines 146–152).

//Reviewer 2 comment:

- Line 136: PAMPs (*pl*).

//Author reply:

As the reviewer pointed out, we corrected the word “PAMP” to “PAMPs” in the revised manuscript (line 142).

//Reviewer 2 comment:

- Line 254: Are the GSH1-GFP-expressing plants in the *gsh1* background? This is not indicated in the figure.

//Author reply:

As the reviewer pointed out, GSH1-GFP was expressed in the *gsh1* mutant background. We have added information on this plant line and revised the sentence in the figure legend (lines 1105–1106).

//Reviewer 2 comment:

- Line 271: “not” is missing?

//Author reply:

As the reviewer pointed out, we corrected the sentence as follows in the revised manuscript (line 307–308): “where the plastid-CFP marker was observed, but chlorophyll autofluorescence was not”

//Reviewer 2 comment:

- Line 305: Does Ref. 53 show this?

//Author reply:

In Ref. 53, Rekhter et al. observed the localization pattern of ICS1-CFP with chlorophyll autofluorescence (Fig. 3A in Ref. 53). Although the Rekhter et al. did not mention the preferential localization of ICS in epidermal small chloroplasts (for what the authors called “plastids”) in the text, Rekhter's Fig. 3A clearly showed strong fluorescent ICS1-CFP signals in epidermal small chloroplasts with chlorophyll compared to in large mesophyll chloroplasts (Rekhter et al., 2019). We have corrected the sentences in the revised manuscript as follows (lines 340–342):

“Interestingly, Rekhter et al. previously provided images showing preferential localization of ICS1/SID2 in small epidermal chloroplasts compared to large mesophyll chloroplasts, although they did not refer to this point in the text⁵³.”

References:

53. Rekhter, D. et al. Isochorismate-derived biosynthesis of the plant stress hormone salicylic acid. *Science* **365**, 498-502 (2019).

//Reviewer 2 comment:

- Line 316: “constitute the immune system” sounds way too strong – please tone down.

//Author reply:

As the reviewer suggested, we toned down the expression and changed the words “cooperatively constitute the immune system” to “contribute to preinvasive immunity” (line 351).

//Reviewer 2 comment:

- English language needs editing throughout the text.

//Author reply:

Almost all parts of our manuscript have been edited by an English language editing service.

//Reviewer 2 comment:

- In my opinion, title and abstract do not clearly convey the conclusions of the work.

//Author reply:

To clearly convey the conclusions, we changed the phrase “motile guardian” to “defense-related motile organelle” in the title, abstract, and main text in the revised manuscript (lines 1, 27, 95).

We have also changed the word “guardians” to “a defense-related organelle” in the revised manuscript (lines 596–597).

We also added Supplementary Fig. 16 in the revised manuscript to visualize the summary of our manuscript.

Finally, we reiterate our thanks to Reviewer 2 for improving our manuscript.

Responses to reviewer 3

//Reviewer 3 comment:

Reviewer #3 (Remarks to the Author):

*The manuscript entitled 'Epidermal chloroplast is a motile guardian equipped with plant immune components' is interesting in that the authors found the requirement of epidermal chloroplasts for plant nonhost resistance to nonadapted *Colletotrichum* fungi. By genetic and cell biological approaches, the authors reached the conclusion that emerged epidermal chloroplasts in response to fungal attempts comprise an additional, likely lower-level, immune system to the PEN2/PEN3 pathway. The previously unrevealed importance of epidermal chloroplasts for plant immunity can clearly attract an interest from many plant scientists. However, the authors should address the below issues to support their claim.*

//Author reply:

Thank you for your critical reading of our manuscript and your insightful suggestions and positive comments, which have helped us to improve our manuscript considerably. Based on the comments provided by the reviewers, we revised our manuscript after performing new experiments. We have addressed all of the reviewers' concerns. Please see our point-by-point responses.

//Reviewer 3 comment:

*1. The authors tested *bak1*, *bik1* and *pepr1 pepr2* mutants for finding an ECR cue, and failed. But, based on the failure to induce ECR by mutant fungi not to form pegs, it is suggested that a mechanical stimulus is likely to induce ECR. Indeed, it was reported that application of a mechanical stimulus induces a subset of plant immune responses (PNAS 95: 8398). Therefore, the authors should test whether mechanical wounding itself can induce ECR.*

//Author reply:

As the reviewer pointed out, Gus-Mayer et al. reported that mechanical stimuli by micromanipulators induce a subset of defense responses in parsley cells (Gus-Mayer et al., 1998 PNAS). In the present study, we determined the rate of the ECR in *Arabidopsis* by counting the number of epidermal pavement cells with or without surface chloroplasts throughout the pathogen-inoculated leaves (please see Methods section). It is technically difficult to uniformly stimulate all epidermal cells throughout the leaf using micromanipulator before ECR analysis. In the alternative method, mechanical stimulus using silicon carbide particles (carborundum), the most effective tool to uniformly wound the epidermal cells on the plant leaf before viral inoculation, could induce plant immune responses by rubbing the entire surface of the leaf (Ogawa et al., 2010). Therefore, we performed carborundum

treatment; however, as in the case of progressive fungal invasion at 3 dpi in New Figs. 1C, New Fig. 8D, and New Fig. 9B, intense carborundum stimuli caused damages to epidermal cells, and we could not quantify the ECR. Furthermore, mild treatment with carborundum throughout the leaf surface did not induce either the ECR or expression of defense-related genes (data not shown). Thus, it is difficult to evaluate the effects of artificial mechanical stimuli on ECR induction.

It is reported that an artificial mechanical stimulus mimicked the penetration attempts of nonadapted *Colletotrichum* fungus (*Corb*) and partially induced penetration resistance in barley, but it was not sufficient to fully induce the penetration resistance triggered by penetration attempt of *Corb* (Kobayashi and Kobayashi, 2013). To clarify the induction of immune response in nonhost *Arabidopsis* against the peg formation-defective *Corb* mutants *cst1* and *pls1*, we additionally analyzed papilla formation in *Arabidopsis* after inoculation with these *Corb* mutants. We found that the *cst1* and *pls1* mutants did not induce this defense response (New Fig. 2E). In contrast, the *Corb sec22* deletion mutant with lower secretion activity triggered full induction of papilla formation (New Fig. 2D), indicating that this *Corb* mutant provides normal or near-normal mechanical stimulus to trigger papilla formation. Thus, the lower levels of ECR induction by the *Corb sec22* deletion mutant indicated the possibility that mechanical stimulus was not the direct trigger of the ECR (New Fig. 2C). We have added the new results of papilla analysis to the manuscript (line 178). However, there still remains a possibility that the slight difference in the degree of fungal progression in the penetration attempt, which is not reflected by papilla formation, is involved in the activation of the ECR. Therefore, we have also added an explanation of this point (lines 180–182).

Reference:

Gus-Mayer et al. Local mechanical stimulation induces components of the pathogen defense response in parsley. *Proc. Natl. Acad. Sci. USA* **95**, 8398-8403 (1998).

Ogawa et al. Transient increase in salicylic acid and its glucose conjugates after wounding in *Arabidopsis* leaves. *Plant Biotech.* **27**, 205-209 (2010)

Kobayashi and Kobayashi. Microwounding is a pivotal factor for the induction of actin-dependent penetration resistance against fungal attack. *Planta* **237**, 1187-1198 (2013).

//Reviewer 3 comment:

2. The authors found that *Cfio* induces some defense-related genes in *pen2* mutant but not in WT. They explained this with a threshold concept that ECR level is low in WT to induce those genes. However, in Fig. 1C, ECR level in WT at 3 dpi is comparable to that in *pen2* mutant at 1 and 2 dpi. Then, those genes are induced at 3 dpi in WT?

//Author reply:

In accordance with the reviewer's comment, we added new quantification data on the expression of defense-related genes in the wild-type plant at 72 hpi (3 dpi) (New Supplementary Fig. 2). As shown in the new results, most of the tested genes were not induced even after 72 hpi of *Cfio*. However, the additionally identified nonadapted fungus *Cnym*, which showed a somewhat higher entry rate into wild-type plants with higher ECR induction than *Cfio* (New Fig. 8), induced moderate defense-related gene expression in the wild-type plant at 72 hpi (New Supplementary Fig. 14). These results suggest a correlation between ECR activation and induced expression of these defense-related genes in the wild-type plant, as well as in *pen2* mutants. Our hypothesis is that the threshold of the ECR in each epidermal cell is low compared to the induction of these genes; however, the population of ECR-activated epidermal cells throughout the pathogen-inoculated leaf gradually increases during the incubation with pathogens, which might activate defense-related gene expression. It remains to be elucidated why defense-related gene expression was not induced in the wild-type at 72 hpi of *Cfio*. Quick activation of the ECR may be important to determine this aspect. To clearly convey this in the manuscript, we added an additional explanation and revised the relevant sentences (lines 503–510).

//Reviewer 3 comment:

3. *Since there are many multiple mutants, it is understandable for the authors to omit some controls (WT and respective single mutants) in most figures showing fungal entry rates and gene expression. However, for readability, the authors should show all these controls.*

//Author reply:

In accordance with the reviewer's comment, we added wild-type and respective single mutants as controls in New Fig. 4A, 4E, New Fig. 7A, 7B, 7D, New Supplementary Fig. 10, and New Supplementary Fig. 11.

//Reviewer 3 comment:

4. *In some mutants shown in Supp Fig 11, fungal entry rates of Cfio and Csia reached almost 90%. These mutants can also allow more penetration of an adapted fungus. How is the susceptibility of Chig in those mutants? Indeed, ECR to Chig is significantly elevated in pen2 mutant in Fig. 1C.*

//Author reply:

As the reviewer pointed out, for the multiple mutants with 90% or higher entry rates of *Cfio* and *Csia*, we further checked the susceptibility of the plants to adapted *Chig* (New Supplementary Fig. 12). As the reviewer expected, all the tested multiple mutants also showed enhanced susceptibility to *Chig*. Thus, we added the following sentence in the revised manuscript (lines 352–354).

“On the other hand, many multiple mutant plants showed enhanced susceptibility to *Chig*, suggesting that these factors also contribute to basal resistance against adapted *Colletotrichum* fungus (Supplementary Fig. 12).”

//Reviewer 3 comment:

5. The authors argued that the ECR system might work as a lower-level mechanism to the PEN2/PEN3 system. Then, the authors should consider why plants have this additional immune system and when the PEN2/PEN3 pathway can be disrupted during plant immune responses.

//Author reply:

We agree with the reviewer; we also think it is important to consider why plants have the ECR as an additional immune system, in addition to when the higher-layer preinvasive defenses (including the PEN2/PEN3 pathway) can be disrupted during plant immune responses. As for the question of disrupting the PEN2/PEN3 pathway, we hypothesized that a more aggressive nonadapted fungus exists in nature that can effectively suppress the PEN2/PEN3 pathway. In this case, the corresponding nonadapted fungus induced high levels of the ECR in wild-type plants. To address this hypothesis, we searched and identified the additional nonadapted *Colletotrichum* fungus *Cnym*, which showed a somewhat higher entry rate into wild-type *Arabidopsis* than *Cfio* (New Fig. 8). Importantly, *Cnym* induced high levels of the ECR in the wild-type plant; consistently, the *CHUP1ox* and *chup1* single mutants permitted higher entry of *Cnym* compared to wild-type plants (New Fig. 8C, 8D). Please see the new paragraph on the experiments using *Cnym* (lines 379–403). We have also revised some sentences in the relevant parts of the Discussion section (lines 472–474, 493–501). As for the question of why plants have this additional immune system, we consider that the ECR acts as a backup system when the first layered defense is overcome by particular nonadapted pathogens, which guarantees plant survival.

//Reviewer 3 comment:

6. In lines 434-436 on page 20, maybe 'increased' instead of 'decreased'?

//Author reply:

As the reviewer pointed out, we corrected the word “decreased” to “increased” (line 541).

We have also added Supplementary Fig. 16 in the revised manuscript to visualize the summary of our manuscript.

Finally, we reiterate our thanks to Reviewer 3 for improving our manuscript.

Responses to reviewer 4

//Reviewer 4 comment:

Reviewer #4 (Remarks to the Author):

The manuscript „Epidermal chloroplast is a motile guardian equipped with plant immune components“ reports on novel discovery that chloroplast movement contributes to leaf epidermal immunity to fungal invasion. This observation is particularly relevant for pen2-immuno-compromised Arabidopsis genotypes in interaction with directly cell wall penetrating fungi. Cell biological studies are supported by genetic evidence that chloroplast movement/recruitment to the cell surface actively contributes to pre-invasive immunity. Little is known about the contribution of chloroplasts in general to plant immunity. This is even more true for small epidermal chloroplasts, which are generally under-explored in epidermal plant cell biology. I therefore think the contribution gives interesting and perhaps surprising insight into a new function of chloroplasts in plant immunity. I enjoyed reading and found data largely convincing. However, I have several points to mention, which, I feel, would contribute to further improve the study/manuscript and make conclusions more precise.

//Author reply:

Thank you for your critical reading of our manuscript and your insightful suggestions and positive comments, which have helped us to improve our manuscript considerably. Based on the comments provided by the reviewers, we revised our manuscript after performing new experiments. We have addressed all of the reviewers' concerns. Please see our point-by-point responses.

//Reviewer 4 comment:

General comments:

I wonder whether the authors have some evidence, that ECR would be also of importance in a wilt type background because basically all results are in the pen2 background. I see that this is perhaps necessary to have an immune-compromised background to see the patho-phenotypes, but there is a certain concern, that we are looking a pen2 pleiotropic effects, which must not come to action in a more natural scenario. Another way to show this would be testing other pen mutants.

//Author reply:

We agree with the reviewer; we also think it is important to consider that the ECR would also be important in a wild-type background. All the results of the experiments using *Cfio*, *Csia* and *Corb* suggested that the ECR is an additional immune system when the higher-layer preinvasive defenses, such as PEN2-related immunity, are ineffective. To address the role of the ECR under natural conditions in *Arabidopsis*, we searched and identified additional nonadapted *Colletotrichum*

fungus *Cnym*, which showed a somewhat higher entry rate into wild-type *Arabidopsis* than *Cfio* (New Fig. 8). Importantly, *Cnym* induced high levels of the ECR; consistently, the *CHUP1ox* and *chup1* single mutants permitted more higher entry of *Cnym* compared to wild-type plants (New Fig. 8C, 8D). Please see the new paragraph on the experiments using *Cnym* (lines 379–403). We also revised the sentence in the relevant part of the Discussion section as follows (lines 470–474).

“We concluded that the ECR is programmed to be activated when higher-layer preinvasive NHR, such as the PEN2-related pathway, is not effective; this is because a specific nonadapted fungus, which partly overcomes higher-layer preinvasive defense(s), induces the greatest ECR in the wild-type *Arabidopsis* background (Fig. 8).”

//Reviewer 4 comment:

The CHUP mutants affect cell entry of parasitic fungi but hardly affect expression of defense genes. It therefore remains unclear how ECR could contribute to pre-invasive immunity. This could be explained by diverse mechanisms, which would be more or less specific for the actual host immune response. The conclusions that epidermal chloroplasts act in immunity by positioning immune (from the abstract “motile guardians specifically accommodating immune regulators in plant epidermis”) components during the host response is therefore tempting. However, there is actually no direct evidence for this? I believe, to prove this, you would need genetic evidence, that CAS, GSH1 and EDS5 act genetically redundantly with CHUP. This, however, appears not to be the case when looking at Figure 5E. Therefore I am not fully convinced that ECR is functionally linked to or acts “cooperatively” with the other components introduced here (GSH; EDS5, EDR1, CAS).

//Author reply:

By statistically testing some defense-related genes induced by *Cfio* inoculation, we found that there were significant decreases in the expression as a result of CHUP1 overexpression (New Fig. 4E). Therefore, we added the words “significant decrease” and deleted the words “despite non-significance” in the revised sentence (lines 244–247). However, as the reviewer suggested, the effect of ECR impairment on the *Cfio*-induced expression of these genes is still weak. As the reviewer also mentioned here, we consider that the main and direct role of the ECR for plant immunity is a regulatory role for properly positioning the immune components located on epidermal chloroplasts (GSH1, EDS5, CAS, and as-yet-unknown components) in preinvasive NHR.

To further address the link between the ECR and the immune components GSH1, EDS5, and CAS, we additionally generated fluorescent marker plant lines of these immune components in the *CHUP1ox* and *chup1* backgrounds (New Fig. 6). We found that their localization patterns strongly correlated with epidermal chloroplasts and the ECR (New Fig. 5, New Fig. 6). Thus, the ECR clearly

supports drastic changes in the subcellular locations of GSH1, EDS5, and CAS in response to fungal inoculation. We added a new paragraph explaining the relationship between GSH1, EDS5, and CAS locations and the ECR (lines 299–304).

We agree with the reviewer; we also think that the ECR had somewhat additive effects with GSH1, EDS5, and CAS on the NHR against *Corb* (New Fig. 7D). *Corb* showed much lower invasion ability into multiple mutants of nonhost *Arabidopsis* than *Cfio* and *Csia* (New Fig. 7D, New Supplementary Fig. 11). Therefore, we speculated that epidermal chloroplasts also carry other components engaged in additional immune pathway(s), which are effective against *Corb* in highly incompatible *Arabidopsis-Corb* interactions. To address the functional link between the ECR and GSH1, EDS5, and CAS in plant immunity, we performed new experiments using the additionally identified fungus *Csia* (low-invasive strain), which showed lower invasion ability than *Csia*, but showed sufficiently higher invasion ability than *Corb* (New Fig. 7D, New Supplementary Fig. 11). *Csia* (low-invasive strain) induced low levels of the ECR in the wild-type plant, but sufficiently induced the ECR in the *pen2* mutant (New Supplementary Fig. 13). Importantly, *gsh1*, *eds5*, *cas* and ECR-deficient mutations showed redundant effects on NHR against *Csia* (low-invasive strain) (New Fig. 7D). Therefore, we added a new paragraph on *Arabidopsis* NHR against *Csia* (low-invasive strain) (lines 356–377). We also revised the sentence in the relevant part of the Discussion section as follows (lines 543–547).

“More importantly, we showed that the ECR-deficient mutation acted redundantly with *gsh1*, *eds5*, and *cas* mutations in the preinvasive NHR against the low-invasive *Csia* strain (Fig. 7D lower). These results suggest that the movement of these immune components via the ECR is a key factor for epidermis-specific antifungal NHR (Supplementary Fig. 16).”

//Reviewer 4 comment:

I think the use of statistical testing is a bit selective over the entire data set. I guess the manuscript would profit from general ANOVA and post hoc testing in all data panels or from other statistical methods with correction for multiple testings.

//Author reply:

We performed statistical tests (ANOVA and post hoc Tukey HSD or Dunnett's tests) in all bar graphs shown in all figures and revised the Methods section accordingly (lines 783–788).

//Reviewer 4 comment:

wonder whether authors demonstrated ECR by any other means than chloroplast autofluorescence, which, in the way it was recorded, could be also autofluorescence of substances other than chlorophyll.

//Author reply:

We moved Supplementary Fig. 8 to New Supplementary Fig. 1 and changed the image (+ *Cfio*) to a new one with a control (+ DW). In these images, we detected epidermal chloroplasts using fluorescent plastid marker in addition to chlorophyll autofluorescence. We also added the following sentence in the manuscript (lines 112–114): “This phenomenon was also observed in a transgenic plant line expressing the plastid-cyan fluorescent protein (CFP) marker (Supplementary Fig. 1).”

//Reviewer 4 comment:

Other comments:

Line 23: ECR-fixed: I wonder whether this is a good term? Perhaps consider constitutively activated ECR

//Author reply:

We performed a new experiment on the ECR of the *chup1* mutant in response to *Cfio* inoculation at a high concentration (three times) (New Fig. 3D, left). Statistical analysis showed no difference in the percentage of epidermal pavement cells with surface chloroplasts with or without fungal inoculation, but additional quantification revealed that the number of surface chloroplasts in the epidermis increased to some extent (New Fig. 3D, right). Thus, we conclude that the *chup1* mutant still has the ability to slightly move epidermal chloroplasts. *CHUP1ox* plants were defective in the movement of epidermal chloroplasts (ECR) (Fig. 3B, New Fig. 3C). In contrast, the *chup1* mutant constitutively fixed a specified population of epidermal chloroplasts at the surface area and restricts their movement, thereby resulting in an ECR-deficient phenotype (Fig. 3B, New Fig. 3C). In terms of impairments of CHUP1-associated moving ability of epidermal chloroplasts in the ECR, both mutants commonly exhibit impaired ECR. To clarify these points, we have added new results and additional explanations in the revised manuscript (lines 202–209). We also removed the word “ECR-fixed” in the revised manuscript.

//Reviewer 4 comment:

Line 27: what do you mean with accommodating? Please be more precise here!

//Author reply:

As the reviewer suggested, we changed the word “accommodate” to “position” in the revised manuscript (line 27) and legend in New Fig. 5 (line 1103).

//Reviewer 4 comment:

Line 48: perhaps consider mentioning what type of transporter PEN3 is.

//Author reply:

As the reviewer suggested, we added an explanation about PEN3 and revised the manuscript as follows (lines 49–50). “ATP binding cassette transporter PEN3”.

//Reviewer 4 comment:

Line 59: chloroplasts do not produce calcium: Do you mean release?

//Author reply:

As the reviewer suggested, we corrected the sentence in the revised manuscript (lines 60–61), as follows: “Secondary messengers such as reactive oxygen species (ROS) and calcium (Ca²⁺) are produced and released by chloroplasts, respectively”.

//Reviewer 4 comment:

Line 92: ECR does not enhance nonhost resistance but is required for, or involved in, or functions in.

//Author reply:

As the reviewer suggested, we have changed the phrase “to enhance” to “involved in” in the revised manuscript (line 95).

//Reviewer 4 comment:

Line 163: I am actually not sure whether you can conclude this based on a correlation. The delta_sec22 mutant certainly has also a virulence defect and therefore less fungal progression may simply cause less ECR response.

//Author reply:

It is reported that an artificial mechanical stimulus mimicked penetration attempt of nonadapted *Colletotrichum* fungus (*Corb*) and partially induced penetration resistance in barley, but it was not sufficient to fully induce penetration resistance triggered by the penetration attempt of *Corb* (Kobayashi and Kobayashi, 2013).

To clarify the induction of immune response in nonhost *Arabidopsis* against the peg formation-defective *Corb* mutants *cst1* and *pls1*, we additionally analyzed papilla formation in *Arabidopsis* after inoculation with these *Corb* mutants. We found that *cst1* and *pls1* mutants did not induce this defense response (New Fig. 2E). In contrast, the *Corb sec22* deletion mutant with lower secretion activity triggered full induction of papilla formation (New Fig. 2D), indicating that this *Corb* mutant provides normal or near-normal mechanical stimulus to trigger papilla formation. Thus, the lower levels of ECR

induction by the *Corb sec22* deletion mutant indicated the possibility that mechanical stimulus was not the direct trigger of the ECR (New Fig. 2C). We have added the new results of papilla analysis to the manuscript (line 178). However, there still remains a possibility that the slight difference in the degree of fungal progression in the penetration attempt, which is not reflected in papilla formation, is involved in ECR activation. Therefore, we have also added an explanation of this point (lines 180–182).

Reference:

Kobayashi and Kobayashi. Microwounding is a pivotal factor for the induction of actin-dependent penetration resistance against fungal attack. *Planta* **237**, 1187-1198 (2013).

//Reviewer 4 comment:

Figure 5a:

The localization of GSH1, EDS5 etc. on small epidermal chloroplasts and at stomules looks specific to me. However, I wonder whether we are looking at z-stacks here. If not, please add the entire z-stack from the same cell so that the reader can distinguish specific localization from cytoplasmic background (CAM35S-over-expressed?). Also mention in the figure legend, which promoter was used for expression.

//Author reply:

As the reviewer suggested, we added Z-stack images and movies of each fluorescent marker line (New Supplementary Fig. 8, New Supplementary Movie 2). These data further support the specific localization of GSH1, EDS5, and CAS on small epidermal chloroplasts and stomules. We expressed GSH1-GFP under its own promoter in the *gsh1-1* mutant, while EDS5-sfGFP and CAS-sfGFP were expressed under the CaMV 35S promoter in wild-type plants. We have added an explanation about promoters in the legend in New Fig. 5 (lines 1106-1107).

//Reviewer 4 comment:

Line 316: To me, data point to additive rather than cooperative functions.

//Author reply:

In the corresponding part of the manuscript, we changed the phrase “cooperatively constitute the immune system” to “contribute to preinvasive immunity” (line 351). As the reviewer pointed out, the ECR had somewhat additive effects with GSH1, EDS5, and CAS on NHR against *Corb*, which showed much lower invasion ability into multiple mutants of nonhost *Arabidopsis* than *Cfio* and *Csia* (New Fig. 7C, 7D, New Supplementary Fig. 11). However, as mentioned above, we additionally identified

Csia (low-invasive strain), which showed lower invasion ability than *Csia* (New Fig. 7D, New Supplementary Fig. 11) and low ECR induction in wild-type plants (New Supplementary Fig. 13). Importantly, *gsh1*, *eds5*, *cas* and ECR-deficient mutations showed redundant effects on NHR against *Csia* (low-invasive strain) (New Fig. 7D), as mentioned above.

//Reviewer 4 comment:

Line 340: What do you mean by guide? The wording appears a bit inappropriate to me.

//Author reply:

As the reviewer suggested, we changed the phrase “function as a guide for” to “support” in the revised manuscript (line 429).

//Reviewer 4 comment:

Line 361: Nuclear movement is CHUP-dependent not ECR dependent: Please be more precise, at least in the results part.

//Author reply:

In a previous study, it was reported that CHUP1 regulates the photorelocation movements of both the chloroplast and nucleus; however, CHUP1-GFP was observed only on the epidermal and mesophyll chloroplast envelopes, but not on the nuclear envelope (Higa et al., 2014). Along with our data on the pathogen-induced ECR, these results indicated that nuclear movements in response to light and pathogenic fungi depend on chloroplast movement, which is directly regulated by CHUP1. Therefore, we added the following sentence in the Results section (lines 450–453).

“Since CHUP1 localizes only to the epidermal and mesophyll chloroplast envelopes and not to the nuclear envelope⁵⁴, it is suggested that CHUP1-dependent ECR repression results in the inhibition of nuclear movement to the epidermal surface with *Cfio* inoculation.”

Reference:

54. Higa, T. et al. Actin-dependent plastid movement is required for motive force generation in directional nuclear movement in plants. *Proc. Natl. Acad. Sci. USA* **111**, 4327-4331 (2014).

//Reviewer 4 comment:

Line 388: “penetration-specific”. This could also be mechanical triggers, right?

//Author reply:

To state more precisely, we changed the phrase “penetration-specific” to “penetration stage-specific”

in the revised manuscript (line 482).

//Reviewer 4 comment:

Line 390-91: I do not agree. I rather see an indirect effect of the sec22 mutation. Also, if an effector would be recognized we would expect it to trigger a hypersensitive reaction, which is a hallmark of plant effector-triggered immunity.

//Author reply:

The data clearly showed the correlation between fungal effector secretion at the penetration pore and plant ECR (New Fig. 2B, 2C). Thus, we lined up the fungal effector as one possible candidate with other candidates for triggers of the ECR (lines 484–491). As the reviewer pointed out, the recognition of pathogen effectors usually induces a hypersensitive response, which is often accompanied by localized cell death. We detected strong cell death of the plant epidermis in response to *Colletotrichum* and *Magnaporthe* fungi (New Fig. 4F, New Fig. 9D), but there was no evidence that these cell deaths were triggered by effector recognition. Therefore, we have added the following sentence to the revised manuscript (line 485–486): “, although the recognition of pathogen effectors usually induces a hypersensitive response.”

However, as the reviewer pointed out, there still remains a possibility that the slight difference in the degree of fungal progression in the penetration attempt, which is not reflected in papilla formation, is involved in ECR activation. Therefore, we have also added an explanation on this point (lines 180-182).

//Reviewer 4 comment:

Line 403-405: To my understanding, gene expression and ECR were recorded at different points in time and should therefore not compared to each other.

//Author reply:

We quantified the ECR at 1, 2, and 3 dpi (24, 48, and 72 hpi) (New Fig. 1C) and analyzed the *Cfio*-induced expression of defense-related genes at 24 hpi (New Supplementary Fig. 2). Additionally, we added new quantification data on *Cfio*-induced gene expression at 72 hpi in the revised manuscript (New Supplementary Fig. 2). Thus, these data can be compared.

//Reviewer 4 comment:

Line 445: I think there is a lot of literature that shows nucleus translocation to sites of fungal attack. Please consider discussing this.

//Author reply:

In accordance with the reviewer's comment, we added the new sentences describing organelle translocation to the site of fungal attack in the Discussion section according to the following references (lines 564–568):

Reference:

68. Gross et al. Translocation of cytoplasm and nucleus to fungal penetration sites is associated with depolymerization of microtubules and defence gene activation in infected, cultured parsley cells. *EMBO J.* **12**, 1735-1744 (1993).

69. Heath et al. Plant nuclear migrations as indicators of critical interactions between resistant or susceptible cowpea epidermal cells and invasion hyphae of the cowpea rust fungus. *New Phytol.* **135**, 689-700 (1997).

70. Shan and Goodwin. Reorganization of filamentous actin in *Nicotiana benthamiana* leaf epidermal cells inoculated with *Colletotrichum destructivum* and *Colletotrichum graminicola*. *Int. J. Plant Sci.* **166**, 31-39 (2005).

71. Opalski et al. The receptor-like MLO protein and the RAC/ROP family G-protein RACB modulate actin reorganization in barley attacked by the biotrophic powdery mildew fungus *Blumeria graminis* f.sp. *hordei*. *Plant J.* **41**, 291-303 (2005).

//Reviewer 4 comment:

Line 457-458: Your data do not show transport by but localization at stromules.

//Author reply:

In accordance with the reviewer's comment, we have toned down the expression and revised the sentences as follows:

“Therefore, our findings suggest that the ECR may at least partially regulate the localization of *Arabidopsis* immune-related proteins via stromules.” (lines 312–314)

“Therefore, our data indicate that stromules may potentially transport immune-related *Arabidopsis* proteins.” (lines 579–580)

We also changed the word “transport” to “intracellular traffic” in the revised manuscript (line 592).

//Reviewer 4 comment:

Line 468: There is only one immune system, to which ECR might contribute.

//Author reply:

As the reviewer pointed out, we changed the phrase “immune system” to “responses of atypical small chloroplasts” in the revised manuscript (line 590).

//Reviewer 4 comment:

Line 474: What do you mean with guardians. Actually, you do not show intrinsic function of chloroplasts themselves but rather of other components that hitchhike on them. I personally would appreciate a clearer and less projecting wording of your findings.

//Author reply:

As the reviewer suggested, to clearly convey the conclusions, we changed the phrase “motile guardian” to “defense-related motile organelle” in the title, abstract, and main text in the revised manuscript (lines 1, 27, 95). We have also changed the word “guardians” to “defense-related organelle” in the revised manuscript (line 596-597).

We have added Supplementary Fig. 16 in the revised manuscript to visualize the summary of our manuscript.

Finally, we reiterate our thanks to Reviewer 4 for improving our manuscript.

REVIEWERS' COMMENTS

Reviewer #1 (Remarks to the Author):

Authors have addressed most of the comments raised in the previous version.

Reviewer #2 (Remarks to the Author):

The authors have made a commendable effort to tackle the reviewers' questions and concerns in this revised version of their manuscript. I only have some minor comments:

- English is remarkably improved throughout the manuscript, but some sentences/expressions sound a bit strange/unspecific (but please note that I am not a native English speaker, hence take this remark with a grain of salt): see lines 205 ("A more fungal inoculum"), 357 ("many multiple mutant plants"), 391 ("slightly severe lesions"), 416-417 and 504 ("slightly invaded"), 515-516 ("even slightly overcome").

- I disagree with the authors in their statement that their findings suggest that the ECR may at least partially regulate the localization of immune-related proteins via stromules (lines 316-318 and 584-585). Any stromal protein in the chloroplast will be detectable in stromules – chloroplast-targeted free GFP is -, and therefore this localization is by far not sufficient to support this idea. Of course, the hypothesis proposed by the authors is valid, but I would suggest that it is toned down.

- Lines 458-459: is the "equipped with many NHR components" required in this sentence?

Reviewer #3 (Remarks to the Author):

In the revised manuscript, the authors well-addressed all concerns that I raised. With this revision, the authors' claim is more convincingly supported. Therefore, I feel that this revised manuscript can contribute to understanding the importance of epidermal chloroplasts for plant immunity, and is publishable in the journal.

Reviewer #4 (Remarks to the Author):

Thank for the revision of the manuscript. You carefully adressed my most important concerns and I found the contribution now more round and overall convincing.

Responses to reviewer 1:

//Reviewer 1 comment:

Reviewer #1 (Remarks to the Author):

Authors have addressed most of the comments raised in the previous version.

//Author reply:

We reiterate our thanks to Reviewer 1 for improving our manuscript.

Responses to reviewer 2

//Reviewer 2 comment:

Reviewer #2 (Remarks to the Author):

The authors have made a commendable effort to tackle the reviewers' questions and concerns in this revised version of their manuscript. I only have some minor comments:

//Author reply:

Thank you for your critical reading of our manuscript and positive comments. We have revised our manuscript in accordance with your minor comments. Please see our point-by-point responses.

//Reviewer 2 comment:

English is remarkably improved throughout the manuscript, but some sentences/expressions sound a bit strange/unspecific (but please note that I am not a native English speaker, hence take this remark with a grain of salt): see lines 205 (“A more fungal inoculum”), 357 (“many multiple mutant plants”), 391 (“slightly severe lesions”), 416-417 and 504 (“slightly invaded”), 515-516 (“even slightly overcome”).

//Author reply:

As the reviewer pointed out, we revised the manuscript as follows:

- i) We changed the phrase “A more fungal inoculum” to “An inoculum with high concentration of fungal conidia” (Lines 203–204).
- ii) We changed the phrase “many multiple mutant plants” to “many plants with multiple mutations” (Line 355).
- iii) We changed the phrase “slightly severe lesions” to “small visible lesions” (Lines 389–390).
- iv) We changed the sentence “*M. oryzae* slightly invaded the *pen2* mutant” to “*M. oryzae* invaded the *pen2* mutant to a slight degree” (Lines 414–415).
- v) We changed the sentence “*Cfio* slightly invaded Col-0” to “*Cfio* invaded Col-0 to a slight degree” (Line 502).
- vi) We changed the sentence “fungal pathogens even slightly overcome higher-layer preinvasive defense(s)” to “fungal pathogens overcome higher-layer preinvasive defense(s) even in a slight degree” (Lines 513–514).

//Reviewer 2 comment:

I disagree with the authors in their statement that their findings suggest that the ECR may at least partially regulate the localization of immune-related proteins via stromules (lines 316-318 and 584-

585). *Any stromal protein in the chloroplast will be detectable in stromules – chloroplast-targeted free GFP is -, and therefore this localization is by far not sufficient to support this idea. Of course, the hypothesis proposed by the authors is valid, but I would suggest that it is toned down.*

//Author reply:

In accordance with the reviewer's comment, we toned down the expression and revised the sentences in the revised manuscript as follows: "Therefore, the stromules possibly link to the regulation of the localization of *Arabidopsis* immune-related proteins via ECR." (Lines 315–316), "Therefore, our data indicate the potential link between immune-related *Arabidopsis* proteins and stromules." (Lines 583–584)

//Reviewer 2 comment:

Lines 458-459: is the "equipped with many NHR components" required in this sentence?

//Author reply:

We agree with the reviewer and removed the following sentence in the revised manuscript (Line 456): "equipped with many NHR components"

Finally, we reiterate our thanks to Reviewer 2 for improving our manuscript.

Responses to reviewer 3

//Reviewer 3 comment:

Reviewer #3 (Remarks to the Author):

In the revised manuscript, the authors well-addressed all concerns that I raised. With this revision, the authors' claim is more convincingly supported. Therefore, I feel that this revised manuscript can contribute to understanding the importance of epidermal chloroplasts for plant immunity, and is publishable in the journal.

//Author reply:

We reiterate our thanks to Reviewer 3 for improving our manuscript.

Responses to reviewer 4

//Reviewer 4 comment:

Reviewer #4 (Remarks to the Author):

Thank for the revision of the manuscript. You carefully adressed my most important concerns and I found the contribution now more round and overall convincing.

//Author reply:

We reiterate our thanks to Reviewer 4 for improving our manuscript.